# SelectLLM – Calibrating LLMs for Selective Prediction: Balancing Coverage and Risk

## Abstract

Despite the impressive capabilities of large language models (LLMs), their outputs often exhibit inconsistent correctness and unreliable factual accuracy. In high-stakes domains, overconfident yet incorrect predictions can lead to serious consequences, highlighting the need for robust uncertainty estimation. To address this, we introduce SelectLLM, an end-to-end method designed to enhance the ability of LLMs to recognize and express uncertainty effectively. By integrating selective prediction into finetuning, SelectLLM optimizes model performance over the covered domain, achieving a more balanced trade-off between predictive coverage and utility. Experimental results on TriviaQA, CommonsenseQA and MedConceptsQA show that SelectLLM significantly outperforms standard baselines, improving abstention behaviour while maintaining high accuracy.

## 1 Introduction

Large language models (LLMs) have rapidly become foundational components in natural language processing (NLP), driving progress across a wide range of tasks – from open-ended generation to complex reasoning. Despite their huge progress and impressive capabilities, LLMs still frequently produce outputs with varying levels of correctness and factual accuracy. A core challenge in deploying these models in real-world settings lies in balancing accuracy with calibrated confidence. While high accuracy remains a primary goal, it is equally critical for models to recognize and signal their own uncertainty, particularly in high-stakes scenarios such as healthcare (Busch et al., 2025; Denecke et al., 2024), finance (Yoo, 2024; Nie et al., 2024), and law (El Hamdani et al., 2024; Colombo et al., 2024). Overconfident incorrect responses can be significantly more harmful than abstentions or cautious, low-confidence responses. To address this, we leverage confidence modeling to enable selective prediction, allowing the system to abstain from answering when uncertainty is high (Wen et al., 2024), thereby trading off *coverage* for *reliability*. This trade-off is especially important in safety-critical applications or decision-support systems, where deferring uncertain cases to a human or fallback system is preferable to propagating potentially erroneous outputs. In this paper, we introduce a principled approach to enhancing safety of an LLM that allows a model to abstain from making a prediction when it is uncertain, thereby reducing the risk of harmful or misleading outputs. However, abstention introduces a secondary trade-off: while conservative behavior can reduce risk, excessive abstention diminishes the utility of the model by forgoing opportunities where correct responses are feasible. A model that abstains too frequently may be safe but ultimately useless. For example, in the "needle in the-haystack" benchmark, LLMs become more uncertain when given the "nonexistent" option, even when capable of providing correct answers (Kim et al., 2025). This highlights the challenge of balancing risk with utility (coverage): optimizing both the correctness of answers and the number of answered questions.

We formalize this challenge as a risk-coverage trade-off and categorize model outputs into four distinct cases following the previous literature (Stengel-Eskin et al., 2024; Cheng et al., 2024), as illustrated in Table 1: ❶ *Accepting a correct answer* — the ideal case, contributing to both utility and reliability; ❷ *Rejecting an incorrect answer* — also desirable, as it avoids unreliable answers; ❸ *Rejecting a correct answer* — suboptimal, reducing the utility of the model; ❹ *Accepting an incorrect answer* — the most harmful case, compromising the accuracy of the model. Our objective is to maximize the occurrence of the first two cases while minimizing the occurrence of the latter two.

To illustrate the risk-coverage trade-off challenge, consider two medical AI assistants designed to help doctors interpret diagnostic test results. Assistant A, optimized solely for utility, studied all diagnostic topics uniformly but lacks the ability to accurately judge when to abstain. Consequently, it sometimes provides incorrect answers with high confidence or unnecessarily abstains even when it could have answered correctly. In contrast, Assistant B explicitly accounts for the risk-coverage trade-off by carefully distinguishing between cases it can confidently address and those it should avoid. When faced with ambiguous diagnostic cases, Assistant B appropriately abstains, whereas in clear-cut cases that Assistant A might wrongly skip, Assistant B reliably provides accurate answers. Consequently, Assistant B achieves the best average diagnostic performance, as illustrated in Figure 1.

To address this challenge, we propose a novel method, called SelectLLM, that explicitly produces confidence estimates and incorporates the task of confidence estimation into its training objectives. SelectLLM assigns confidence scores to questions rather than to generated answers, thereby quantifying the reliability of the LLM's response to specific queries independent from the multiple alternative answers generated. Questions can be classified into two categories based on a confidence threshold: those with confidence above a given threshold (covered by the model) and those below the threshold (not covered). Within the covered set of questions, we further distinguish between the questions the model is confident in answering correctly and those it confidently identifies as beyond its capability, corresponding to the first and second cases mentioned previously.

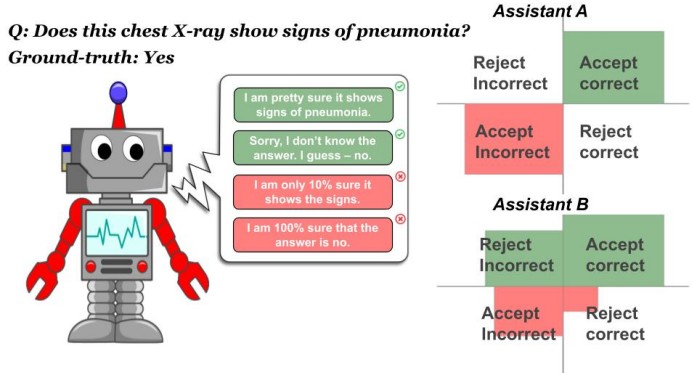

Figure 1: Illustration of risk-coverage trade-off. Given a question, Assistant A (base LLM), optimized solely for utility, often produces incorrect answers due to overconfidence. In contrast, Assistant B (with SelectLLM), which explicitly accounts for the risk–coverage trade-off, recognizes its limitations and abstains when uncertain. As a result, it avoids more errors and achieves better performance on diagnostic tasks.

SelectLLM is based on a well-trained LLM and jointly trains (fine-tunes the first and trains the second) two heads (shown in Figure 2): ❶ a *decoding head*, corresponding to the original LLM output layer for autoregressive token generation; ❷ a *selection head*, outputting a confidence score for the question. This two-head design is motivated by the known calibration deficiencies of trained LLMs. In a well-calibrated model, the decoding head's next-token probabilities could be used directly for confidence estimation. However, LLMs often exhibit overconfidence or underconfidence, making it necessary to learn a separate abstention signal. The selection head is explicitly optimized to improve the risk–coverage trade-off, allowing the model to balance utility with reliability. Our contributions are summarized as follows:

- We introduce SelectLLM, which incorporates risk–coverage trade-off control into the LLM training stage. It combines **Direct Preference Optimization (DPO)** (Rafailov et al., 2023) with confidence estimation to improve the risk-coverage trade-off;

- We **construct three high-quality benchmarks** for DPO fine-tuning based on open-sourced Question-and-Answer datasets, and conduct extensive experiments on **seven baselines with three different LLMs**, demonstrating that SelectLLM significantly outperforms state-of-the-art baselines in terms of risk and coverage metrics;

- We validate the confidence scores produced by SelectLLM by comparing their distribution to scores derived from the tone and phrasing of the generated responses, demonstrating that SelectLLM can natively output reliable confidence estimates for its predictions **without relying on any external models**.

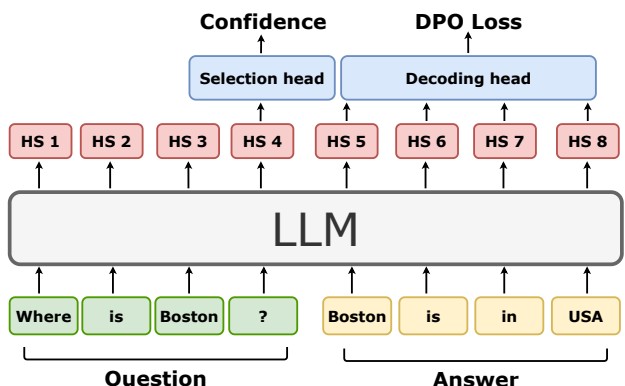

*Figure 2:* **Overview of SelectLLM.** *Given a question–answer input pair, the underlying LLM processes the full sequence and produces a hidden state (HS) for each token. The selection head operates on the hidden state corresponding to the last token of the question to estimate a confidence score for abstention; while the decoding head uses the answer-related hidden states to compute the DPO loss for LLM fine-tuning. This dual-head design enables SelectLLM to jointly optimize for utility and accuracy.*

*Table 1: Four cases of the answer to a question: "In which branch of the arts does Allegra Kent work?".*

|  | **Accept** (high confidence) | **Reject** (low confidence) |
|---|---|---|
| **Correct** | Allegra Kent is a ballet dancer. She worked as a principal dancer with the New York City Ballet. | I'm not entirely certain, but I think Allegra Kent might be involved in ballet. |
| **Incorrect** | Allegra Kent is a renowned opera singer who performed in major productions throughout Europe. | I'm not really sure, but maybe Allegra Kent is a painter? |

## 2 RELATED WORK

**Uncertainty Quantification in LLMs.** Uncertainty estimation for large language models (LLMs) spans several complementary paradigms. and generally falls into two categories: (i) black-box approaches and (ii) white-box approaches. Black-box methods include verbalized uncertainty, where models are prompted to express confidence in natural language (Tanneru et al., 2023; Yona et al., 2024; Wei et al., 2023; Lin et al., 2023; Huang et al., 2024), and sampling-based methods, which estimate predictive uncertainty from variability across multiple generations (Cole et al., 2023; Ji et al., 2024; Xiong et al., 2023). White-box approaches, in contrast, exploit model internals such as token-level probabilities, calibration of log-likelihoods, or hidden-state diagnostics to produce confidence scores. Related work includes TokenSAR (Duan et al., 2023), P(True) (Kadavath et al., 2022) and Semantic Entropy (Kuhn et al., 2023). While many of these techniques primarily serve to identify uncertain predictions and guide abstention, there is also a growing line of work on uncertainty-aware training, where uncertainty estimates inform parameter updates (Krishnan et al., 2024; Niu et al., 2024; Yang et al., 2023b). Our approach builds on these advances by directly incorporating selective prediction objectives into fine-tuning.

**Alignment and Confidence in LLMs.** Efforts to align LLMs with human preference, such as Proximal Policy Optimization (PPO) (Schulman et al., 2017) and Direct Preference Optimization (DPO) (Rafailov et al., 2023), adjust model parameters to encourage desired behaviours. Kang et al. (2024) proposed conservative reward modeling to encourage LLMs to be more cautious in their predictions, which relates to our objective of selective prediction. Piché et al. (2024) introduced self-restraint fine-tuning, aiming to increase model confidence when appropriate while reducing overconfidence. Recent works such as (Stengel-Eskin et al., 2024) and (Cheng et al., 2024) utilize DPO to align LLMs with human preference to guide the model to answer questions it knows and to avoid answering questions it does not know.

**Selective Prediction in LLMs.** Selective prediction has a rich history in machine learning (Fumera & Roli, 2002; Wiener & El-Yaniv, 2015; Cortes et al., 2016a;b), and has recently been extended to LLMs (Yoshikawa & Okazaki, 2023; Lee et al., 2024; Srinivasan et al., 2024; Yang et al.,

2023a). However, none of these LLM-related works incorporates selective coverage into model training. SelectiveNet (Geifman & El-Yaniv, 2019) provides a foundational framework for selective classification in deep networks. Our work extends this idea to the generative setting of LLMs, which poses unique challenges. SelectLLM differs from prior frameworks such as SelectiveNet in several critical ways. While SelectiveNet targets classification and regression, SelectLLM is designed for sequence generation. To enable this, we introduce a new module that embeds the generated sequence before passing it to a confidence head, enabling reliable abstention decisions for natural language outputs. Moreover, SelectiveNet employs three heads—reward, selection, and auxiliary—to encourage shared representation learning. In contrast, SelectLLM adds only a single selection head $g(\cdot)$ to the original LLM and fine-tunes the entire framework to align with human preferences. This design enables SelectLLM to balance generation quality, prediction accuracy, and selective abstention, offering a principled framework for calibrated and trustworthy language generation.

In summary, by synthesizing advances from uncertainty quantification, fine-tuning, and selective prediction, SelectLLM introduces a principled framework that jointly optimizes predictive performance and uncertainty estimation, a contribution of particular significance for high-stakes applications.

## 3 PROBLEM FORMULATION

We define *coverage* as the proportion of questions for which the model is confident enough to provide an answer:

$$\text{coverage} = \frac{1}{n} \sum_{i=1}^{n} (1 - a_i),$$

where $n$ is the total number of questions, $a_i = 1$ if the model abstains on the $i$th question and $a_i = 0$ otherwise. While *risk* is defined as the error rate over the set of answered questions:

$$\text{risk} = \frac{\sum_{i=1}^{n} \mathbb{1}(\hat{y}_i \notin \mathcal{Y}_i \wedge a_i = 0)}{\sum_{i=1}^{n} (1 - a_i)},$$

where $\hat{y}_i$ is the model's output, $\mathcal{Y}_i$ is the set of correct answers for the $i$th question.

The goal is to ensure that LLMs can reliably estimate their predictive confidence and abstain when uncertainty is high, while also minimizing unnecessary abstentions to retain practical utility. Our approach is built on Direct Preference Optimization (DPO) (Rafailov et al., 2023), a human preference alignment method that fine-tunes language models using pairwise comparisons of answers without the need to explicitly model a reward function.

DPO (Rafailov et al., 2023) is a human preference alignment method that fine-tunes language models using comparisons of pairs of answers without the need to explicitly model a reward function. Specifically, in the **training** stage, we are given (1) a dataset $\mathcal{X} = \{x_1, x_2, \ldots, x_n\}$, where each $x_i$ is a question posed to the model; (2) a corresponding set of human preference annotations $\mathcal{P} = \{(y_{i,+}, y_{i,-})\}$, where $y_{i,+}$ and $y_{i,-}$ denote the preferred and rejected answers to question $x_i$, respectively; and (3) a predefined coverage rate $0 < c < 1$, which represents the target proportion of questions for which the user expects the model to provide confident answers. Our goal is to maximize the likelihood of human-preferred answers relative to rejected ones given the coverage constraint $c$, yielding a fine-tuned model $M_{select}$ and a selection head $g(\cdot)$ which outputs a confidence score $c_i$ indicating the model's confidence in answering a specific question $x_i$.

In the **inference** stage, given (1) a dataset of input questions, $\mathcal{X} = \{x_1, x_2, \ldots, x_n\}$, where each $x_i$ is a question; and (2) a trained model $M_{select}$ and its selection head $g(\cdot)$, the model produces (1) a set of LLM-generated answers, $\hat{\mathcal{Y}} = \{\hat{y}_1, \hat{y}_2, \ldots, \hat{y}_n\}$, where each $\hat{y}_i$ is the model's answer to $x_i$; and (2) a set of confidence scores, $\mathcal{C} = \{\text{conf}_1, \text{conf}_2, \ldots, \text{conf}_n\}$, where each $\text{conf}_i$ represents the model's confidence that it can answer question $x_i$ correctly.

Given the model's answer to a question, together with its confidence score to answer the question, the model abstains when its confidence score $\text{conf}_i$ is below a given threshold $\tau$. More formally, the abstention decision for question $x_i$ is defined as $a_i = \begin{cases} 1 & \text{if } \text{conf}_i < \tau \\ 0 & \text{otherwise} \end{cases}$.

# 4 SELECTLLM

Our proposed method SelectLLM enhances pre-trained LLMs by introducing an additional head that explicitly estimates the model's confidence in answering a given question correctly. This selection head is trained or fine-tuned jointly with the base model. Specifically, given a pre-trained LLM $\pi_\theta$, we augment it with a selection head $g(\cdot)$, which outputs a confidence score conf $\in (0, 1)$.

Unlike traditional confidence estimation methods that rely on token-level probabilities, our selection head operates on the last-layer hidden state of the final token in the input question. This design ensures that confidence estimation is based solely on the model and the input question.

## 4.1 SELECTION HEAD ARCHITECTURE

The selection head is implemented as a lightweight two-layer multilayer perceptron (MLP) that maps the final hidden representation of the language model to a scalar confidence score. Concretely, the hidden state of the last token is first transformed using a linear projection from the model dimension $d_{\text{model}}$ to a 512-dimensional intermediate space, followed by a ReLU nonlinearity and a dropout layer with probability $p = 0.1$. A second linear layer maps this 512-dimensional vector to a single logit, which is subsequently passed through a sigmoid activation to produce a confidence value in the range $(0, 1)$. This compact architecture adds negligible overhead while enabling the model to learn a calibrated selection policy.

The additional cost introduced by the selection head is negligible relative to the backbone computation. First, during training, the gradient updates for the large-scale language model (with billions of parameters) dominate the total compute cost, while the selection head contributes only a tiny number of additional parameters and operations, resulting in virtually no change in wall-clock time. Second, during inference, the selection head operates solely on the hidden state computed in the prefill stage, either in parallel with or immediately before generation. Consequently, it adds only a single lightweight matrix multiplication, causing no noticeable increase in latency.

## 4.2 LOSS FUNCTION

The loss function of SelectLLM combines the **DPO loss**, which aligns the model's outputs with human preferences, and the **Select loss**, which manages the risk–coverage trade-off.

The **DPO loss** aligns the model's outputs with human preferences without requiring explicit reward modeling or reinforcement learning. Given a dataset of human preferences $\mathcal{P} = \{(x_i, y_{i,+}, y_{i,-})\}$, where $y_{i,+}$ is the preferred response and $y_{i,-}$ is the rejected response to question $x_i$, the DPO loss is defined as:

$$L_{\text{DPO}}(\pi_\theta, \pi_{\text{ref}}) = -\mathbb{E}_{(x,y_+,y_-)\sim\mathcal{P}} \left[ \log \sigma \left( \beta \log \frac{\pi_\theta(y_+ \mid x)}{\pi_{\text{ref}}(y_+ \mid x)} - \beta \log \frac{\pi_\theta(y_- \mid x)}{\pi_{\text{ref}}(y_- \mid x)} \right) \right] \quad (1)$$

where:

- $\pi_\theta$ is the LLM we want to fine-tune.
- $\pi_{\text{ref}}$ is a reference model, usually a frozen version of the original pre-trained language model.
- $\sigma$ is the sigmoid function.
- $\beta$ is a hyperparameter that controls the amount of divergence from the reference model $\pi_{\text{ref}}$.

Building on Section 3, we define the empirical selective risk for LLM fine-tuning as:

$$\hat{r} = \frac{1}{n} \sum_{i=1}^{n} (g(h_i) \cdot L_{\text{DPO}}) \quad (2)$$

where $h_i$ denotes the hidden state of the last token in the question, $g(h_i) \in [0, 1]$ is the selection function that quantifies the model's confidence for the given question.

Notably, since the original DPO loss only boosts the margin between the chosen answer and the rejected answer, it may simultaneously decrease the probabilities of both chosen and rejected answers, compared to the reference model, which is not desirable (Rafailov et al., 2024; Feng et al., 2024; Pal

et al., 2024). Therefore, we define a marginal function measuring the difference in the probabilities between the answers of the fine-tuned model and the reference model, which is defined as follows:

$$w(y) = \beta \left( \log \pi_\theta(y) - \log \pi_{\text{ref}}(y) \right) \tag{3}$$

where $\beta$ is a hyper-parameter, and $\pi_\theta$, $\pi_{\text{ref}}$ follow the same definitions as in the DPO loss.

Then we define the risk for generating chosen and rejected answers using Equation 3:

$$\ell(\pi_\theta, \pi_{\text{ref}}, y) = \begin{cases} \log \sigma \left( \max(0, -w(y)) \right) & \text{if } y \in y_+ \\ \log \sigma \left( \max(0, w(y)) \right) & \text{if } y \in y_- \end{cases}$$

Mathematically, these are not the same as DPO. DPO encourages $w(y_+) > w(y_-)$; while this additional risk term encourages $w(y_+) \geq 0$ (do not degrade chosen probability below reference) and $w(y_-) \leq 0$ (do not increase rejected probability above reference). The intuition behind this risk is as follows: a penalty is applied if the fine-tuned model assigns a lower probability to chosen answers than the reference model, or a higher probability to rejected answers.

Building on the above, we define a modified empirical selective risk as follows:

$$\hat{r}_\ell(\pi_\theta, \pi_{\text{ref}}, g) = \frac{1}{n} \sum_{i=1}^{n} ((1 - w_+ - w_-) \cdot L_{\text{DPO}} + w_+ \cdot \ell(\pi_\theta, \pi_{\text{ref}}, y_{i,+}) + w_- \cdot \ell(\pi_\theta, \pi_{\text{ref}}, y_{i,-})) \cdot g(h_i) \tag{4}$$

where $w_+$ and $w_-$ are hyper-parameters defined by the users. In the appendix, we include an ablation study to demonstrate the effectiveness of the two additional terms ( $\ell(\pi_\theta, y_{i,+})$ and $\ell(\pi_\theta, y_{i,-})$).

The **Select loss** aims to minimize the selective risk while maintaining a predefined coverage level $c$. Formally, the Select objective is given by:

$$L_{\text{Select}} = \hat{r} + \lambda \cdot \Psi(c - \hat{\phi}(g)) \tag{5}$$

where $\hat{\phi}(g) = \frac{1}{n} \sum_{i=1}^{n} g(h_i)$ is the empirical coverage, $\lambda > 0$ is a regularization parameter, and $\Psi(a) = \max(0, a)^2$ penalizes deviations from the target coverage rate $c$ defined by the user.

Finally, the **Combined loss** is defined as a weighted sum of the Select loss and the fine-tuning loss:

$$L_{\text{Combined}} = \alpha \cdot L_{\text{Select}} + (1 - \alpha) \cdot L_{\text{DPO}} \tag{6}$$

where $\alpha \in [0, 1]$ balances the weight of the two objectives. Following Geifman & El-Yaniv (2017), we set $\alpha = 0.5$ without hyperparameter tuning in all experiments.

If we do not incorporate the Select loss, the model may produce outputs aligned with human preferences but lack effective confidence calibration, which could result in excessive abstention or incorrect responses overly confident. The use of the original DPO loss, $L_{\text{DPO}}$, is also essential to optimizing SelectLLM. Since the selection head is initialized randomly, without $L_{\text{DPO}}$, SelectLLM will focus on a fraction $c$ of the training set, before accurate low level features are constructed. In such a case, SelectLLM will tend to overfit to the wrong subset of the training set. The $L_{\text{DPO}}$ exposes the SelectLLM model to all training instances throughout the training process. Thus, integrating both losses ensures that the model achieves a balanced performance – producing high-quality, preference-aligned outputs while maintaining optimal coverage through calibrated confidence estimation.

## 5 EXPERIMENTS

In this section, we first compare SelectLLM against seven baseline models on the TriviaQA (Joshi et al., 2017) and CommonsenseQA (Talmor et al., 2018) benchmarks, two widely used datasets for evaluating open-domain question-answering systems. We then demonstrate SelectLLM's ability to generalize across domains by fine-tuning on CommonsenseQA and testing on TriviaQA. Next, we validate the confidence scores produced by SelectLLM, followed by an ablation study to assess the impact of the reward loss terms and the coverage–risk trade-off.

## 5.1 EXPERIMENTAL SETUP

We use Llama-3.1-8B-Instruct[1](Grattafiori et al., 2024), Mistral-7B-Instruct-v0.2[2](Jiang et al., 2023) and Qwen2.5-14B-Instruct[3](Yang et al., 2025) in the experiments as the base models. We use QLoRA (Dettmers et al., 2023) with rank 16 to train all the models. For comparison, we use *base* (LLM without finetuning), *LACIE* (Stengel-Eskin et al., 2024) (DPO-based finetuning), *LARS* (Yaldiz et al., 2024) (uses a well-trained score function), *MARS* (Bakman et al., 2024) (uses a QA evaluator model), *TokenSAR* (Duan et al., 2023) (uses a sentence similarity model), *P(True)* (Kadavath et al., 2022) (a self-check method) and *Semantic Entropy (SE)* (Kuhn et al., 2023) (uses token probabilities) as our baselines. For all models, we report average performance across 5 seeds. We perform all the LLM fine-tuning on one A100-40GB GPU.

**Metrics.** Across all the experiments, we report the following evaluation metrics: the number of true positives **(TP)**, the number of true negatives **(TN)**, **Precision**, **Recall**, and **Coverage**. We also include the **TRUTH** metric introduced in Cheng et al. (2024), defined as the sum of TP and TN, which captures the number of correctly accepted and correctly abstained responses. Because the test dataset contains 1,000 samples, the upper bound of TRUTH is 1,000. As there are no ground-truth or reference confidence scores provided for each question, we cannot report AUROC or ECE scores.

For score-based methods (SelectLLM, LARS, MARS, TokenSAR, P(True), and SE), we tune a threshold on the validation set to maximize the TRUTH metric and then apply the same threshold to the test set for abstention. For non-score-based methods (base and LACIE), we use a rule-based evaluation strategy: a response is accepted as long as the model provides an answer and is rejected only if the model explicitly refuses or states that it does not know.

**Datasets.** We use the TriviaQA (Joshi et al., 2017), CommonsenseQA (Talmor et al., 2018), and MedConceptsQA (Shoham & Rappoport, 2024) datasets. Following Stengel-Eskin et al. (2024), for TriviaQA we randomly sample 10,000, 1,000, and 1,000 questions for the training, validation, and test sets, respectively. For CommonsenseQA, we randomly sample 8,000, 1,000, and 1,000 questions for the training, validation, and test sets, respectively. For MedConceptsQA, which is used solely for evaluation, we randomly sample 1,000 questions each for the validation and test sets.

To construct the chosen/rejected pairs used for LACIE and SelectLLM fine-tuning, we first augment each dataset with model-generated answers and their associated confidence scores. Specifically, we use the base models mentioned above to generate an answer for each question and then employ DeepSeek-v3 (Liu et al., 2024) to assign a confidence score based on the tone and phrasing of the generated response. We refer to this score as *tone-confidence*. The prompt provided to DeepSeek-v3 is: *"Rate how confident the response appears based solely on its tone and phrasing."*

We set a confidence threshold of 0.7: answers with scores above this threshold are accepted, while those below are rejected. If no correct answer exceeds the threshold, we default to a generic response—*"I don't know the answer."*—as the chosen answer. Such fallback responses occur in roughly 30% of the fine-tuning dataset. All remaining answers to the same question are treated as rejected. Finally, we construct the fine-tuning pairs for both LACIE and SelectLLM by sampling one chosen and one rejected answer for each question.

## 5.2 IN-DISTRIBUTION PERFORMANCE

We conduct experiments on the TriviaQA and CommonsenQA datasets. As shown in Table 2&3, our method SelectLLM, consistently and substantially improves model truthfulness and precision across all three language models. It achieves the highest TRUTH score by a significant margin in every experiment—for instance, reaching 752.0 with Llama-3.1 compared to the base model's 601.7. This strong performance is primarily driven by its unique strength in correctly abstaining from providing an answer, as evidenced by its leading True Negative (TN) values (e.g., 230.3 for Mistral-7B on TriviaQA and 142.6 on CommonsenseQA). In contrast, all other score-based methods (LARS, MARS, TokenSAR, P(True), SE) fail to provide a reliable confidence score, since their low TN counts and

---

[1]https://huggingface.co/meta-llama/Llama-3.1-8B-Instruct

[2]https://huggingface.co/mistralai/Mistral-7B-Instruct-v0.2

[3]https://huggingface.co/Qwen/Qwen2.5-14B-Instruct

*Table 2: TriviaQA performance. ↑ indicates the higher the better, and ↓ indicates the lower the better. **The TN value for both the base and LACIE is 0.0 (with a corresponding Recall of 1.0), since they do not abstain from any answers.***

| Model | TP ↑ | TN ↑ | TRUTH ↑ | Precision ↑ | Recall ↑ | Coverage (%) |
|---|---|---|---|---|---|---|
| *Llama-3.1-8B-Instruct* | | | | | | |
| base | $\mathbf{601.7}_{\pm 2.3}$ | $0.0_{\pm 0.0}$ | $601.7_{\pm 2.3}$ | $0.602_{\pm 0.002}$ | $\mathbf{1.000}_{\pm 0.000}$ | $100.0_{\pm 0.0}$ |
| LARS | $579.3_{\pm 3.7}$ | $45.2_{\pm 4.0}$ | $624.2_{\pm 6.5}$ | $0.627_{\pm 0.018}$ | $0.949_{\pm 0.005}$ | $92.4_{\pm 8.8}$ |
| MARS | $556.2_{\pm 8.9}$ | $57.1_{\pm 2.4}$ | $613.4_{\pm 7.6}$ | $0.626_{\pm 0.017}$ | $0.912_{\pm 0.015}$ | $88.9_{\pm 9.9}$ |
| TokenSAR | $559.2_{\pm 9.3}$ | $62.3_{\pm 6.2}$ | $621.1_{\pm 7.9}$ | $0.630_{\pm 0.006}$ | $0.916_{\pm 0.022}$ | $88.7_{\pm 14.6}$ |
| P(True) | $565.6_{\pm 2.1}$ | $54.8_{\pm 4.1}$ | $621.9_{\pm 5.4}$ | $0.622_{\pm 0.014}$ | $0.965_{\pm 0.015}$ | $94.7_{\pm 3.7}$ |
| SE | $589.5_{\pm 7.4}$ | $32.1_{\pm 5.8}$ | $619.3_{\pm 7.5}$ | $0.627_{\pm 0.010}$ | $0.926_{\pm 0.011}$ | $90.1_{\pm 12.8}$ |
| LACIE (DPO) | $579.3_{\pm 23.6}$ | $0.0_{\pm 0.0}$ | $579.3_{\pm 23.6}$ | $0.579_{\pm 0.024}$ | $\mathbf{1.000}_{\pm 0.000}$ | $100.0_{\pm 0.0}$ |
| SelectLLM ($c = 0.75$) | $582.0_{\pm 19.7}$ | $\mathbf{170.0}_{\pm 25.2}$ | $\mathbf{752.0}_{\pm 2.6}$ | $\mathbf{0.773}_{\pm 0.015}$ | $0.884_{\pm 0.021}$ | $75.96_{\pm 3.63}$ |
| *Mistral-7B-Instruct-v0.2* | | | | | | |
| base | $\mathbf{598.3}_{\pm 4.0}$ | $0.0_{\pm 0.0}$ | $598.3_{\pm 9.0}$ | $0.598_{\pm 0.009}$ | $\mathbf{1.000}_{\pm 0.000}$ | $100.0_{\pm 0.0}$ |
| LARS | $587.4_{\pm 7.5}$ | $48.2_{\pm 3.4}$ | $635.3_{\pm 8.2}$ | $0.626_{\pm 0.010}$ | $0.977_{\pm 0.008}$ | $93.8_{\pm 12.9}$ |
| MARS | $558.5_{\pm 8.1}$ | $40.2_{\pm 4.2}$ | $598.1_{\pm 2.9}$ | $0.608_{\pm 0.013}$ | $0.928_{\pm 0.010}$ | $91.7_{\pm 4.7}$ |
| TokenSAR | $529.4_{\pm 8.7}$ | $61.2_{\pm 2.5}$ | $590.9_{\pm 4.8}$ | $0.610_{\pm 0.012}$ | $0.880_{\pm 0.016}$ | $86.7_{\pm 11.9}$ |
| P(True) | $532.8_{\pm 10.9}$ | $81.2_{\pm 5.1}$ | $613.1_{\pm 6.4}$ | $0.626_{\pm 0.009}$ | $0.885_{\pm 0.015}$ | $85.0_{\pm 8.3}$ |
| SE | $582.3_{\pm 8.3}$ | $33.7_{\pm 6.0}$ | $615.3_{\pm 3.6}$ | $0.614_{\pm 0.020}$ | $0.968_{\pm 0.009}$ | $94.8_{\pm 18.0}$ |
| LACIE (DPO) | $568.4_{\pm 3.4}$ | $0.0_{\pm 0.0}$ | $568.4_{\pm 7.4}$ | $0.568_{\pm 0.007}$ | $\mathbf{1.000}_{\pm 0.000}$ | $100.0_{\pm 0.0}$ |
| SelectLLM ($c = 0.70$) | $522.0_{\pm 19.9}$ | $\mathbf{230.3}_{\pm 24.7}$ | $\mathbf{752.3}_{\pm 12.3}$ | $\mathbf{0.741}_{\pm 0.019}$ | $0.891_{\pm 0.039}$ | $70.87_{\pm 4.21}$ |
| *Qwen2.5-14B-Instruct* | | | | | | |
| base | $\mathbf{636.2}_{\pm 10.7}$ | $0.0_{\pm 0.0}$ | $636.2_{\pm 10.7}$ | $0.636_{\pm 0.011}$ | $\mathbf{1.000}_{\pm 0.000}$ | $100.0_{\pm 0.0}$ |
| LARS | $624.0_{\pm 6.4}$ | $17.1_{\pm 4.2}$ | $641.2_{\pm 2.0}$ | $0.643_{\pm 0.016}$ | $0.981_{\pm 0.008}$ | $97.1_{\pm 3.7}$ |
| MARS | $605.7_{\pm 7.2}$ | $27.2_{\pm 5.1}$ | $632.1_{\pm 7.7}$ | $0.642_{\pm 0.011}$ | $0.951_{\pm 0.011}$ | $94.2_{\pm 9.5}$ |
| TokenSAR | $580.4_{\pm 2.3}$ | $72.2_{\pm 11.8}$ | $652.6_{\pm 3.6}$ | $0.665_{\pm 0.015}$ | $0.912_{\pm 0.012}$ | $87.2_{\pm 7.4}$ |
| P(True) | $613.1_{\pm 11.1}$ | $34.7_{\pm 6.5}$ | $647.2_{\pm 13.9}$ | $0.650_{\pm 0.020}$ | $0.964_{\pm 0.013}$ | $94.3_{\pm 12.5}$ |
| SE | $624.2_{\pm 9.5}$ | $30.3_{\pm 2.4}$ | $654.7_{\pm 5.8}$ | $0.651_{\pm 0.011}$ | $0.981_{\pm 0.008}$ | $95.8_{\pm 14.6}$ |
| LACIE (DPO) | $646.7_{\pm 3.3}$ | $0.0_{\pm 0.0}$ | $646.7_{\pm 3.3}$ | $0.647_{\pm 0.003}$ | $\mathbf{1.000}_{\pm 0.000}$ | $100.0_{\pm 0.0}$ |
| SelectLLM ($c = 0.80$) | $599.5_{\pm 24.3}$ | $\mathbf{141.8}_{\pm 20.2}$ | $\mathbf{741.3}_{\pm 9.8}$ | $\mathbf{0.745}_{\pm 0.021}$ | $0.919_{\pm 0.027}$ | $80.55_{\pm 5.14}$ |

only marginal precision gains over the base model demonstrate an inability to effectively identify and filter out incorrect answers. We further analyze the confidence scores generated by SelectLLM in Section 5.5.

Consequently, when SelectLLM does generate a response, its reliability is much higher, reflected in its top-ranking Precision scores (e.g., 0.745 for Qwen2.5 on TriviaQA vs. the base model's 0.636). This enhanced precision comes with a deliberate sacrifice of lower Coverage and Recall, as SelectLLM strategically answers fewer questions to avoid making errors. This demonstrates its effectiveness for applications where accuracy is more critical than providing an answer to every query.

## 5.3 OUT-OF-DISTRIBUTION GENERALIZATION

To further assess the generalizability of SelectLLM, we evaluate its performance on out-of-distribution (OOD) datasets. Specifically, the tested models are fine-tuned on CommonsenseQA, without any additional fine-tuning on the test datasets – TriviaQA and MedConceptsQA. The evaluation results are reported in Table 7&5. The results demonstrate that the learned abstention ability is transferable to OOD datasets. While the base and LACIE (DPO) models, which lack an abstention mechanism, are forced to answer every question, resulting in a True Negative (TN) of 0.0 and a low Precision, SelectLLM successfully transfers its learned skill of abstaining from uncertain queries to the unseen domains. This is clearly evidenced by its high TN counts: 74.0 on TriviaQA and a remarkable 172.0 on MedConceptsQA. By correctly identifying and abstaining from these challenging OOD questions, SelectLLM significantly boosts its Precision and surpasses the performance of both the base models and LACIE (DPO). The successful transfer of its capability results in a higher TRUTH score, showing that SelectLLM is not only more reliable in familiar settings but also exhibits robustness and generalizability when faced with novel data.

Table 3: *CommonsenseQA performance.* ↑ *indicates the higher the better, and* ↓ *indicates the lower the better.* **The TN value for both the base and LACIE is 0.0 (with a corresponding Recall of 1.0), since they do not abstain from any answers.**

| Model | TP ↑ | TN ↑ | TRUTH ↑ | Precision ↑ | Recall ↑ | Coverage (%) |
|---|---|---|---|---|---|---|
| *Llama-3.1-8B-Instruct* | | | | | | |
| base | $627.3_{\pm 10.1}$ | $0.0_{\pm 0.0}$ | $627.3_{\pm 10.1}$ | $0.627_{\pm 0.004}$ | $\mathbf{1.000}_{\pm 0.000}$ | $100.0_{\pm 0.0}$ |
| LARS | $575.6_{\pm 4.4}$ | $14.2_{\pm 6.2}$ | $589.0_{\pm 9.1}$ | $0.616_{\pm 0.019}$ | $0.917_{\pm 0.027}$ | $93.4_{\pm 12.8}$ |
| MARS | $567.3_{\pm 7.2}$ | $11.1_{\pm 6.1}$ | $578.8_{\pm 8.9}$ | $0.610_{\pm 0.011}$ | $0.904_{\pm 0.010}$ | $92.9_{\pm 11.0}$ |
| TokenSAR | $554.3_{\pm 7.5}$ | $21.1_{\pm 6.4}$ | $575.7_{\pm 12.5}$ | $0.612_{\pm 0.020}$ | $0.884_{\pm 0.014}$ | $90.6_{\pm 5.5}$ |
| P(True) | $566.1_{\pm 6.9}$ | $13.3_{\pm 5.7}$ | $579.7_{\pm 4.7}$ | $0.611_{\pm 0.013}$ | $0.903_{\pm 0.018}$ | $92.6_{\pm 9.2}$ |
| SE | $559.4_{\pm 7.6}$ | $20.0_{\pm 5.9}$ | $579.1_{\pm 3.4}$ | $0.613_{\pm 0.020}$ | $0.891_{\pm 0.031}$ | $91.2_{\pm 9.9}$ |
| LACIE (DPO) | $\mathbf{733.7}_{\pm 12.2}$ | $0.0_{\pm 0.0}$ | $733.7_{\pm 12.2}$ | $0.734_{\pm 0.012}$ | $\mathbf{1.000}_{\pm 0.000}$ | $100.0_{\pm 0.0}$ |
| SelectLLM ($c = 0.85$) | $697.2_{\pm 23.1}$ | $\mathbf{98.6}_{\pm 22.1}$ | $\mathbf{795.8}_{\pm 11.2}$ | $\mathbf{0.834}_{\pm 0.016}$ | $0.915_{\pm 0.027}$ | $83.28_{\pm 4.09}$ |
| *Mistral-7B-Instruct-v0.2* | | | | | | |
| base | $596.2_{\pm 12.9}$ | $0.0_{\pm 0.0}$ | $596.2_{\pm 10.9}$ | $0.596_{\pm 0.009}$ | $\mathbf{1.000}_{\pm 0.000}$ | $100.0_{\pm 0.0}$ |
| LARS | $595.9_{\pm 8.7}$ | $19.5_{\pm 4.1}$ | $614.5_{\pm 7.3}$ | $0.607_{\pm 0.012}$ | $0.998_{\pm 0.004}$ | $98.0_{\pm 9.3}$ |
| MARS | $582.3_{\pm 7.5}$ | $26.8_{\pm 6.7}$ | $608.1_{\pm 9.5}$ | $0.606_{\pm 0.016}$ | $0.976_{\pm 14.1}$ | $96.0_{\pm 12.4}$ |
| TokenSAR | $571.3_{\pm 6.8}$ | $27.5_{\pm 2.7}$ | $598.2_{\pm 6.2}$ | $0.602_{\pm 0.019}$ | $0.958_{\pm 0.016}$ | $94.8_{\pm 21.9}$ |
| P(True) | $563.6_{\pm 7.4}$ | $51.7_{\pm 5.5}$ | $614.1_{\pm 6.9}$ | $0.614_{\pm 0.010}$ | $0.945_{\pm 18.7}$ | $91.6_{\pm 10.9}$ |
| SE | $579.3_{\pm 11.2}$ | $24.6_{\pm 10.3}$ | $603.9_{\pm 7.1}$ | $0.604_{\pm 0.012}$ | $0.972_{\pm 15.4}$ | $95.9_{\pm 13.3}$ |
| LACIE (DPO) | $603.7_{\pm 9.0}$ | $0.0_{\pm 0.0}$ | $603.7_{\pm 9.0}$ | $0.604_{\pm 9.9}$ | $\mathbf{1.000}_{\pm 0.000}$ | $100.0_{\pm 0.0}$ |
| SelectLLM ($c = 0.80$) | $\mathbf{611.6}_{\pm 29.4}$ | $\mathbf{142.6}_{\pm 27.9}$ | $\mathbf{754.2}_{\pm 10.7}$ | $\mathbf{0.775}_{\pm 0.028}$ | $0.900_{\pm 0.026}$ | $78.8_{\pm 6.43}$ |
| *Qwen2.5-14B-Instruct* | | | | | | |
| base | $800.0_{\pm 12.4}$ | $0.0_{\pm 0.0}$ | $800.0_{\pm 12.4}$ | $0.800_{\pm 0.011}$ | $\mathbf{1.000}_{\pm 0.000}$ | $100.0_{\pm 0.0}$ |
| LARS | $798.2_{\pm 13.8}$ | $19.5_{\pm 7.1}$ | $817.0_{\pm 8.4}$ | $0.815_{\pm 0.010}$ | $0.998_{\pm 0.005}$ | $97.9_{\pm 8.2}$ |
| MARS | $785.8_{\pm 5.9}$ | $52.2_{\pm 15.3}$ | $837.4_{\pm 6.8}$ | $0.841_{\pm 0.011}$ | $0.981_{\pm 0.008}$ | $93.3_{\pm 7.7}$ |
| TokenSAR | $713.6_{\pm 9.3}$ | $62.2_{\pm 2.8}$ | $775.8_{\pm 12.1}$ | $0.838_{\pm 0.022}$ | $0.891_{\pm 0.014}$ | $85.1_{\pm 7.9}$ |
| P(True) | $768.3_{\pm 7.7}$ | $12.0_{\pm 5.2}$ | $780.2_{\pm 7.6}$ | $0.803_{\pm 0.018}$ | $0.960_{\pm 0.010}$ | $95.6_{\pm 10.9}$ |
| SE | $777.3_{\pm 4.5}$ | $41.7_{\pm 9.9}$ | $818.5_{\pm 7.4}$ | $0.830_{\pm 0.010}$ | $0.971_{\pm 0.009}$ | $93.6_{\pm 8.3}$ |
| LACIE (DPO) | $\mathbf{823.7}_{\pm 4.0}$ | $0.0_{\pm 0.0}$ | $823.7_{\pm 4.0}$ | $0.824_{\pm 0.004}$ | $\mathbf{1.000}_{\pm 0.000}$ | $100.0_{\pm 0.0}$ |
| SelectLLM ($c = 0.90$) | $777.4_{\pm 9.0}$ | $\mathbf{68.6}_{\pm 8.7}$ | $\mathbf{846.0}_{\pm 3.0}$ | $\mathbf{0.884}_{\pm 0.011}$ | $0.938_{\pm 0.016}$ | $88.01_{\pm 1.70}$ |

Table 4: *TriviaQA (out-of-distribution) performance.* **The TN value for both the base and LACIE is 0.0 (with a corresponding Recall of 1.0), since they do not abstain from any answers.**

| Model | TP ↑ | TN ↑ | TRUTH ↑ | Precision ↑ | Recall ↑ | Coverage (%) |
|---|---|---|---|---|---|---|
| *Llama-3.1-8B-Instruct* | | | | | | |
| base | $\mathbf{601.7}_{\pm 2.3}$ | $0.0_{\pm 0.0}$ | $601.7_{\pm 2.3}$ | $0.602_{\pm 0.002}$ | $\mathbf{1.000}_{\pm 0.000}$ | $100.0_{\pm 0.0}$ |
| LACIE (DPO) | $579.3_{\pm 23.6}$ | $0.0_{\pm 0.0}$ | $579.3_{\pm 23.6}$ | $0.579_{\pm 0.024}$ | $\mathbf{1.000}_{\pm 0.000}$ | $100.0_{\pm 0.0}$ |
| SelectLLM ($c = 0.85$) | $555.0_{\pm 12.7}$ | $\mathbf{74.0}_{\pm 10.1}$ | $\mathbf{629.0}_{\pm 13.6}$ | $\mathbf{0.626}_{\pm 0.012}$ | $0.933_{\pm 0.011}$ | $86.72_{\pm 3.67}$ |

Table 5: *MedConceptsQA (out-of-distribution) performance.* **The TN value for both the base and DPO is 0.0 (with a corresponding Recall of 1.0), since they do not abstain from any answers.**

| Model | TP ↑ | TN ↑ | TRUTH ↑ | Precision ↑ | Recall ↑ | Coverage (%) |
|---|---|---|---|---|---|---|
| *Llama-3.1-8B-Instruct* | | | | | | |
| base | $319.0_{\pm 5.13}$ | $0.0_{\pm 0.00}$ | $319.0_{\pm 5.13}$ | $0.319_{\pm 0.05}$ | $\mathbf{1.000}_{\pm 0.00}$ | $100.0_{\pm 0.00}$ |
| LACIE (DPO) | $\mathbf{465.0}_{\pm 37.48}$ | $0.0_{\pm 0.00}$ | $465.0_{\pm 37.48}$ | $0.465_{\pm 0.04}$ | $\mathbf{1.000}_{\pm 0.00}$ | $100.0_{\pm 0.00}$ |
| SelectLLM ($c = 0.75$) | $406.7_{\pm 22.23}$ | $\mathbf{172.0}_{\pm 4.89}$ | $\mathbf{578.7}_{\pm 17.62}$ | $\mathbf{0.543}_{\pm 0.03}$ | $0.839_{\pm 0.01}$ | $75.0_{\pm 0.12}$ |

## 5.4 EFFECTIVENESS ON A NON-QA TASK

To further evaluate the effectiveness of SelectLLM on the non-QA tasks with objective ground truth, we conduct additional experiments on GSM8K, a high-quality grade-school math dataset with verifiable ground truth. We used the same data construction and training pipeline as in the QA setting, using `Mistral-7B-Instruct-v0.2` as the base model. The dataset was split into 6,000 training samples, 1,000 validation samples, and 1,000 test samples. The results in Table 6 show that SelectLLM substantially improves precision and overall correctness on math reasoning without any task-specific engineering. This confirms that our proposed method is task-agnostic and extends naturally to domains with objectively measurable correctness.

*Table 6: GSM8K-math (non-QA) performance.* **The TN value for both the base and DPO is 0.0 (with a corresponding Recall of 1.0), since they do not abstain from any answers.**

| Model | TP ↑ | TN ↑ | TRUTH ↑ | Precision ↑ | Recall ↑ | Coverage (%) |
|---|---|---|---|---|---|---|
| *Llama-3.1-8B-Instruct* | | | | | | |
| Base | $482.0_{\pm 3.8}$ | $0.0_{\pm 0.0}$ | $482.0_{\pm 3.8}$ | $0.482_{\pm 0.003}$ | $\mathbf{1.000}_{\pm 0.0}$ | $100.0_{\pm 0.0}\%$ |
| DPO | $551.0_{\pm 12.2}$ | $0.0_{\pm 0.0}$ | $551.0_{\pm 12.2}$ | $0.551_{\pm 0.012}$ | $\mathbf{1.000}_{\pm 0.0}$ | $100.0_{\pm 0.0}\%$ |
| SelectLLM ($c = 0.7$) | $\mathbf{558.0}_{\pm 15.6}$ | $\mathbf{165.0}_{\pm 18.3}$ | $\mathbf{723.0}_{\pm 16.7}$ | $\mathbf{0.811}_{\pm 0.012}$ | $0.922_{\pm 0.018}$ | $70.39_{\pm 1.9}\%$ |

## 5.5 VALIDATION OF SELECTLLM CONFIDENCE SCORES

In this section, we validate the confidence scores generated by SelectLLM by comparing their distribution with the tone-confidence score (referred to Section 5.3) produced by DeepSeek-v3. To visualize these two distributions, we first divide the tone-confidence scores into five bins ([0.2, 0.36], [0.36, 0.52], [0.52, 0.68], [0.68, 0.84], [0.84, 1.00]). Each sample is assigned to a bin based on its tone-confidence score. We then compute the mean tone-confidence and the mean SelectLLM-generated confidence for the samples within each bin.

Figure 3 illustrates a small distribution difference between the confidence scores produced by SelectLLM and the tone-confidence scores generated by DeepSeek-v3 on two datasets. The close alignment of the mean SelectLLM confidence scores with the corresponding tone-confidence scores across all bins demonstrates that the selection head produces meaningful and well-calibrated confidence estimates. This evidence supports the conclusion that SelectLLM can internally and reliably estimate its own prediction confidence, without requiring external reference models.

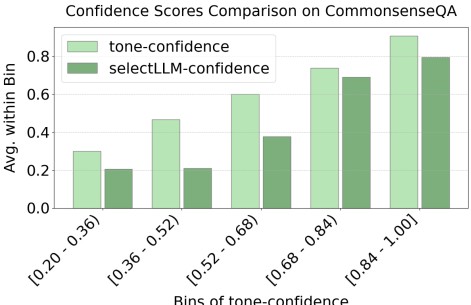 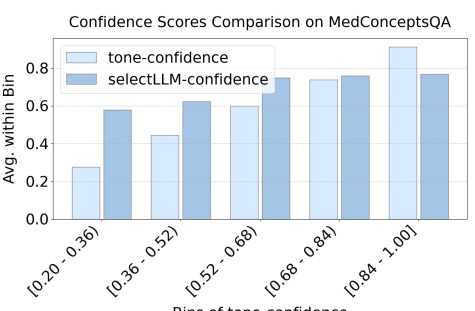

*Figure 3: Distribution Difference between tone-confidence and SelectLLM-confidence for in-distribution (left, CommonsenseQA) and out-of-distribution (right, MedConceptsQA)*

## 6 CONCLUSION

In this paper, we have introduced an alignment-based method, called SelectLLM, that explicitly produces confidence estimates and incorporates the task of confidence estimation into its training objectives. Our extensive empirical evaluations on three QA benchmark datasets, using three different LLMs, demonstrate that SelectLLM consistently achieves better risk-coverage tradeoffs than seven baselines. SelectLLM's notable strengths include superior uncertainty calibration, robust cross-domain generalization, and flexible, tunable performance. These experimental outcomes confirm SelectLLM as an effective and principled solution for enhancing model reliability and practical utility in uncertainty-sensitive, real-world scenarios.

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

## A    QUALITATIVE ANALYSIS

In the following table, we provide the qualitative examples of the base and SelectLLM fine-tuned Llama3.1-8B models, together with the SelectLLM confidence score to the questions. Correct answers are highlighted in green. These examples demonstrate the effectiveness of SelectLLM in two key aspects: it enables the model to express high confidence when it truly knows the answer, and to avoid overconfidence when uncertain. Moreover, SelectLLM produces accurate confidence scores without relying on any external model. This demonstrates SelectLLM 's ability to achieve a better risk–coverage trade-off.

| Question | base (Llama3.1-8B) | SelectLLM |
|---|---|---|
| *While people just throw coins down them now, what originally had a pail to be lowered for it's intended use? Choose the answer from the following choices. (a) garage. (b) utility room. (c) slide. (d) wishing well. (e) garden.* | I'm about 50% confident in my answer. I think the answer is d. I choose this because in a wishing well, a pail or bucket is often lowered down into the well so people can collect water. [Correct & Unconfident] | (d) wishing well. A wishing well is a type of stone-lined well that was originally built with a pail lowered down a rope for people to make a wish and retrieve water. The pail made it possible for people to make a wish and then retrieve water. This is the origin of the wishing well. The answer choice is (d) wishing well. I am 95% sure. [Correct & Confident, SelectLLM-confidence=0.968] |
| *Miss Grady took a stick from Bob because he was playing with it during class. She wanted to make sure that he couldn't get to it so she put it where? Choose the answer from the following choices. (a) desk drawer. (b) kitchen drawer. (c) classroom. (d) pocket. (e) office.* | Miss Grady put the stick where so that Bob could not get to it. In order to do that, she would most likely have put it in a location where she would have control over it. The classroom is where Miss Grady would have the most control over the stick, so that's where she would have put it. Answer: C. [Incorrect & Confident] | (a) or (d) possibly (d) perhaps a pencil or supply box, but (d) a desk drawer or (d) supply cabinet would be most likely. Explanation: Since Miss Grady took the stick from Bob to prevent him from using it, she would likely put it in a secure location where he would not be able to get to it. [Abstention & Unconfident, SelectLLM-confidence=0.051] |
| *What is the description of the medical code S04.019S in ICD10CM? A. Unspecified fracture of unspecified thoracic vertebra, subsequent encounter for fracture with nonunion. B. Terrorism involving fires, conflagration and hot substances, civilian injured, subsequent encounter. C. Injury of optic nerve, unspecified eye, sequela. D. Laceration of abdominal wall with foreign body, epigastric region without penetration into peritoneal cavity, sequela.* | I am 95% confident that my answer is B. Terrorism involving fires, conflagration and hot substances, civilian injured, subsequent encounter. This is because the code S04.019S starts with S̈04ẅhich is the code for terrorism, and the Ö19S̈represents the subsequent encounter for a civilian injured. This makes option B the most likely correct answer. [Incorrect & Confident] | None of the options describe a code that is the same as T32.98 in ICD10CM. The options provided don't seem to be correct. The only way to actually answer this question would be to look at the ICD10CM for the code T32.98. [Abstention & Unconfident, SelectLLM-confidence=0.233] |

## B    EFFECT OF TWO ADDITIONAL LOSS TERMS

We compare the performance of SelectLLM using the Llama-3.1-8B-Instruct model with/without the two reward loss terms mentioned in Equation 4 on TriviaQA benchmark, and report the results in Table 6 ($c = 0.80$). The table shows that integrating the additional reward loss terms significantly boosts overall model performance. Models that incorporate these terms consistently outperform those without, providing empirical validation for the theoretical rationale behind penalizing misalignment relative to the reference model.

## C    ABLATION STUDY

To assess the robustness of SelectLLM, we conducted a sensitivity analysis on five key hyperparameters: the target coverage rate $c$, the loss weighting parameter $\alpha$, the regularization parameter $\lambda$, the learning rate, and batch size. All experiments were performed using Llama-3.1-8B-Instruct on the TriviaQA dataset. As illustrated in Figure 4, SelectLLM demonstrates consistent superiority over the baselines (Base Model and LACIE) across a wide range of hyperparameter configurations. These results confirm that SelectLLM's improvements in the risk-coverage trade-off are robust and not artifacts of narrow hyperparameter tuning.

Table 7: *Ablation Study in terms of the additional loss $\ell(\pi_\theta, y)$ on TriviaQA. ↑ indicates the higher the better, and ↓ indicates the lower the better.*

| Model | TP ↑ | TN ↑ | TRUTH ↑ | Precision ↑ | Recall ↑ | Coverage (%) |
|---|---|---|---|---|---|---|
| *Llama-3.1-8B-Instruct* | | | | | | |
| base | $\textbf{601.7}_{\pm 2.3}$ | $0.0_{\pm 0.0}$ | $601.7_{\pm 2.3}$ | $0.602_{\pm 0.002}$ | $\textbf{1.000}_{\pm 0.000}$ | $100.0_{\pm 0.0}$ |
| SelectLLM w/o $\ell(\pi_\theta, y)$ | $572.1_{\pm 22.3}$ | $123.6_{\pm 21.9}$ | $695.7_{\pm 11.7}$ | $0.704_{\pm 0.019}$ | $0.901_{\pm 0.023}$ | $80.04_{\pm 3.89}$ |
| SelectLLM w/ $\ell(\pi_\theta, y)$ | $599.5_{\pm 24.3}$ | $\textbf{141.8}_{\pm 20.2}$ | $\textbf{741.3}_{\pm 9.8}$ | $\textbf{0.745}_{\pm 0.021}$ | $0.919_{\pm 0.027}$ | $80.55_{\pm 5.14}$ |

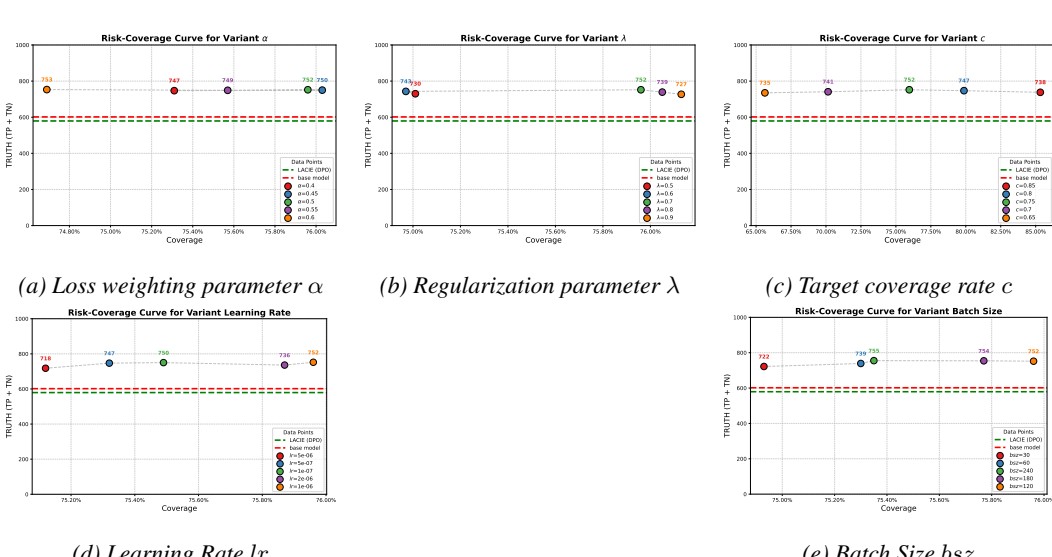

*(a) Loss weighting parameter $\alpha$*     *(b) Regularization parameter $\lambda$*     *(c) Target coverage rate $c$*

*(d) Learning Rate $lr$*             *(e) Batch Size $bsz$*

Figure 4: *Risk-Coverage Curves for Different Hyper-parameters.*

# D   LLM USAGE

This work aims to balance the coverage and accuracy for large language models (LLMs). All the base models tested in this paper are LLMs, including Llama-3.1-8B-Instruct, Mistral-7B-Instruct-v0.2, and Qwen2.5-14B-Instruct. LLMs are also used for language polishing and to improve the paper's readability.

