# OpenReview forum: "SelectLLM – Calibrating LLMs for Selective Prediction: Balancing Coverage and Risk"
_ICLR.cc/2026/Conference — Submitted to ICLR 2026_

### Official Review · Reviewer_tA2u · 2025-10-27

**Soundness:** 2
**Presentation:** 2
**Contribution:** 2
**Rating:** 4
**Confidence:** 3

**Summary:**

This paper introduces SelectLLM, a method for selective prediction that allows LLMs to abstain from answering when uncertain, thereby balancing predictive risk and coverage. SelectLLM employs a dual-head architecture with separate decoding and selection heads. It is jointly fine-tuned using DPO for utility and a custom loss function for calibrated abstention. Extensive experiments on multiple QA benchmarks and LLMs show that SelectLLM outperforms existing baselines.

**Strengths:**

The paper addresses the critical problem of enabling LLMs to abstain when uncertain, which is fundamental for their safe deployment in high-stakes applications. The motivation is well-defined, and the proposed method offers a viable solution. The dual-head architecture, which decouples the generation task from the confidence estimation, is an elegant design choice.

**Weaknesses:**

1. The related work section omits several highly relevant papers on uncertainty quantification and selective prediction for LLMs, such as [1-3]. The paper doesn't provide citations for the LLMs used in the experiments, including Llama-3.1-8B-Instruct, Mistral-7B-Instruct-v0.2, Qwen2.5-14B-Instruct, and DeepSeek-v3.

2. The paper introduces several key hyperparameters without adequate analysis. An ablation study on the modified risk terms in Eq.3 is mentioned (lines 270-271) but is not present in the appendix or main text. This lack of analysis makes it hard to understand why the method works and limits its claimed reliability.

3.  The evaluation of SelectLLM is confined to QA datasets. The paper provides no evidence or  discussion on whether the method generalizes to other tasks, such as summarization and open-ended generation.

**References**:

[1] Uncertainty-aware Language Modeling for Selective Question Answering.  (Yang, et al., Arxiv 2023)

[2] Improving the reliability of large language models by leveraging uncertainty-aware in-context learning.  (Yang, et al., Arxiv 2023)

[3] Uncertainty in language models: assessment through rank-calibration.  (Huang, et al., EMNLP 2024)

**Questions:**

1.  Regarding the tone-confidence metric:

 - How would SelectLLM's performance be affected if a different or less powerful LLM were used to generate the tone-confidence preference labels instead of DeepSeek-v3?
 - Can the authors substantiate the reliability of the tone-confidence used to validate calibration in Section 5.4?

2.  Regarding hyperparameters and implementation details:

 -   How sensitive are the results to the target coverage $c$ and the regularization hyperparameter $\lambda$? Can you provide risk-coverage curves for varying values of these hyperparameters?
 -   What is the specific architecture of the selection head $g(·)$ (e.g., a linear layer, an MLP)? What are the additional training and inference costs associated with this head?
 -   What is the justification for fixing the loss weighting parameter $\alpha$ to 0.5? Is there evidence that this choice is robust across different models and datasets?

3.  Regarding additional experiments and ablations:

 -   Could you provide the ablation study mentioned in the main paper (lines 270-271) that demonstrates the empirical impact of the additional risk terms introduced in Eq.3?
 -   Which component of SelectLLM is most critical to its performance gains? Is it the separate selection head, the explicit coverage constraint $c$ in the loss function, the modified risk formulation, or their combination?
 -   How might SelectLLM be adapted for generative tasks beyond QA? Have you performed any qualitative experiments on tasks like summarization or open-ended generation?

---

> ### Author Response · Authors · 2025-11-25
> **Part 1/N**
>
> We thank the reviewer for the review. Here are our responses:
>
> **W1: The related work section omits several highly relevant papers on uncertainty quantification and selective prediction for LLMs, such as [1-3]. The paper doesn't provide citations for the LLMs used in the experiments, including Llama-3.1-8B-Instruct, Mistral-7B-Instruct-v0.2, Qwen2.5-14B-Instruct, and DeepSeek-v3.**
>
> A1: We thank the reviewer for pointing this out. We have updated the PDF.
>
>  **W2 & Q6: The paper introduces several key hyperparameters without adequate analysis. An ablation study on the modified risk terms in Eq.3 is mentioned (lines 270-271) but is not present in the appendix or main text. This lack of analysis makes it hard to understand why the method works and limits its claimed reliability.**
>
> A2: We thank the reviewer for pointing this out. We have included the missing ablation study in Table 6 in the updated PDF.
>
> **W3 & Q8: The evaluation of SelectLLM is confined to QA datasets. The paper provides no evidence or discussion on whether the method generalizes to other tasks, such as summarization and open-ended generation.**
>
> A3: SelectLLM is intrinsically task-agnostic because it does not rely on hard-coded heuristics or task-specific constraints. The adaptation to tasks like summarization or open-ended generation is straightforward and requires no architectural changes, only a modification to the data construction phase.
>
> We confined our initial evaluation to QA benchmarks (TriviaQA, CommonsenseQA) because they provide objective, non-debatable ground truth for calculating Risk and Coverage. In open-ended generation, "correctness" is often subjective, which would introduce noise into the evaluation of our method's effectiveness.
>
> **Q1: How would SelectLLM's performance be affected if a different or less powerful LLM were used to generate the tone-confidence preference labels instead of DeepSeek-v3?**
>
> A4: The performance of SelectLLM would likely remain robust even with a less powerful annotator, because the external model's role is strictly limited to Tone Analysis, not correctness verification.
>
> As detailed in Section 5.1, we use the external model (DeepSeek-v3) solely to score the "tone and phrasing" of the response. Crucially, the factual correctness of the answer is verified against the dataset's ground truth labels, not by the external LLM. Therefore, the task assigned to the external model is a fundamental NLU task that even smaller, less powerful LLMs (or even rule-based heuristics) can perform with high accuracy.
>
> To empirically validate the robustness of SelectLLM to the choice of the annotator model, we conducted an additional experiment using Qwen3-32B to reconstruct the TriviaQA preference dataset. Qwen3-32B is significantly smaller and less powerful than the DeepSeek-v3 model used in our main experiments. We then retrained the Llama-3.1-8B-Instruct model using SelectLLM on this new dataset.
>
> | Model        | TP (±)        | TN (±)        | TRUTH (±)     | Precision (±) | Recall (±)   | Coverage (%) (±) |
> |--------------|--------------:|--------------:|--------------:|--------------:|--------------:|------------------:|
> | DeepSeek-v3  | 582.0±19.7    | 170.0±25.2    | 752.0±2.6     | 0.773±0.015   | 0.884±0.021   | 75.96±3.63        |
> | Qwen3-32B    | 570.0±12.6    | 183.0±19.4    | 753.0±1.7     | 0.762±0.012   | 0.902±0.018   | 75.33±2.78        |
>
> As shown, the performance is remarkably consistent between the two annotators. The TRUTH scores are nearly identical ($752.0$ vs. $753.0$), and the Coverage differs by less than $0.4\%$. This confirms that SelectLLM does not rely on the superior reasoning capabilities of a massive model like DeepSeek-v3; rather, it only requires the annotator to have basic language understanding to distinguish confident from uncertain tone, a capability well within the reach of smaller models like Qwen3-32B.
>
>
> **Q2: Can the authors substantiate the reliability of the tone-confidence used to validate calibration in Section 5.4?**
>
> A5: To verify that the tone-confidence scores capture an objective linguistic signal rather than an artifact of a specific model, we conducted a consistency check using a significantly less powerful LLM (Qwen3-32B) to annotate the same dataset. We observed a >93% overlap between the binary confidence labels (confident vs. unconfident) generated by this smaller model and the original DeepSeek-v3 labels. This high agreement demonstrates that the "confident tone" is a distinct, recognizable feature of the text that is robust across different evaluators, rather than a subjective or noisy metric.
>
> Additionally, another strong evidence for the reliability of the tone-confidence labels is the performance of SelectLLM itself. If the tone-confidence scores were unreliable or uncorrelated with factual correctness, SelectLLM would fail to distinguish between correct and incorrect answers.

---

> ### Author Response · Authors · 2025-11-25
> **Part 2/N**
>
> **Q3: How sensitive are the results to the target coverage and the regularization hyperparameter? Can you provide risk-coverage curves for varying values of these hyperparameters?**
>
> A6: To evaluate the impact of the target coverage rate $c$ on model performance, we plot a risk-coverage curve with different target coverage rates $c \in \\{0.85, 0.8, 0.75, 0.7, 0.65\\}$ using Llama-3.1-8B-Instruct on the TriviaQA dataset. This figure (Figure 4(c)) is included in our revised PDF. The results demonstrate that while SelectLLM achieves the highest TRUTH score (TP + TN) at $c=0.75$ (the setting used in our main experiments), its performance remains robust across other values. Crucially, SelectLLM consistently and significantly outperforms the DPO baseline across all tested coverage rates.
>
> To evaluate the impact of the coverage regularization term $\lambda$ on model performance, we plot a risk-coverage curve with $\lambda \in \\{0.5, 0.6, 0.7, 0.8, 0.9\\}$ using Llama-3.1-8B-Instruct on the TriviaQA dataset. This figure (Figure 4(b)) is included in our revised PDF.  The results demonstrate that while SelectLLM achieves the highest TRUTH score (TP + TN) at $\lambda=0.7$ (the setting used in our main experiments), its performance remains robust across other values. Crucially, SelectLLM consistently and significantly outperforms the DPO baseline across all tested coverage rates.
>
> **Q4: What is the specific architecture of the selection head (e.g., a linear layer, an MLP)? What are the additional training and inference costs associated with this head?**
>
> A7:
> Architecture:
> The selection head $g(\cdot)$ is a lightweight module (specifically, a 2-layer MLP) that projects the last-layer hidden state of the final question token to a scalar confidence score. It is structurally similar to the value head used in the classification head in standard sequence classification tasks
>
> Computational Costs: The additional costs are negligible for both training and inference.
> 1. Training: The selection head is trained jointly with the LLM backbone. The computational overhead consists only of the forward and backward passes through this single small head. Compared to the heavy matrix multiplications required for the full LLM backpropagation (DPO), this cost is effectively zero.
> 2. Inference: Unlike sampling-based uncertainty methods (e.g., Semantic Entropy, Self-Consistency), which require generating multiple sequences (increasing cost by 5×–10×), or external-model methods (e.g., MARS), which require a second model pass, SelectLLM computes confidence in a single forward pass.
>
> Specifically, the confidence is computed from the pre-fill hidden states of the question prompt.
> This means the confidence score is available before the first token is even generated, adding virtually no latency to the generation process.
>
> **Q5: What is the justification for fixing the loss weighting parameter to 0.5? Is there evidence that this choice is robust across different models and datasets?**
>
> A8: We set $\alpha=0.5$ as following the established theoretical framework of SelectiveNet, which demonstrated that a balanced weighting is effective for joint selection-and-prediction architectures.
>
> We further conduct an ablation study by varying this weighting parameter $\alpha \in \\{0.4, 0.45, 0.5, 0.55, 0.6\\}$ using Llama-3.1-8B-Instruct on the TriviaQA dataset. We find SelectLLM remains robust under different choices of $\alpha$. This result (Figure 4(a)) is included in our revised PDF.

---

> > ### Author Response · Authors · 2025-11-25
> > **Part 3/3**
> >
> > **Q7: Which component of SelectLLM is most critical to its performance gains? Is it the separate selection head, the explicit coverage constraint in the loss function, the modified risk formulation, or their combination?**
> >
> > A9: The performance gains of SelectLLM are driven by the synergistic combination of the separate selection head and the modified risk formulation, rather than any single component in isolation. We break down the criticality of each.
> >
> > The Selection Head: The necessity of a separate selection head is empirically proven by the comparison with LACIE (DPO) in Tables 2 and 3. LACIE utilizes the same DPO fine-tuning but lacks the separate selection head. SelectLLM significantly outperforms LACIE (e.g., TRUTH score 752.0 vs 579.3 on TriviaQA/Llama-3.1), demonstrating that a dedicated module for estimating confidence is superior to relying on the generation head alone.
> >
> > The modified risk (Equation 4) is essential to prevent the "likelihood displacement" issue inherent in standard DPO. As noted in Section 4.1, standard DPO optimizes relative margins and can inadvertently degrade the absolute probability of correct answers. Our ablation study (in Appendix) confirms that the addition of the absolute risk terms ($l(\pi_{\theta}, y_{i,+})$ and $l(\pi_{\theta}, y_{i,-})$) is effective in maintaining the model's ability to confidently identify correct answers.
> >
> > The explicit coverage constraint acts as a necessary regularizer. Without the coverage penalty term $\Psi(c - \hat{\phi}(g))$ in Equation 5, the selection head would trivially minimize risk by abstaining from all predictions (zero coverage).
> >
> > Therefore, removing any of these three components would compromise the system.

---

> ### Comment · Reviewer_tA2u · 2025-11-26
>
> I thank the authors for their response. While some of my concerns have been addressed, several critical issues remain regarding the paper's completeness and clarity.
>
> **1. Regarding W1:**
> Although the authors stated they updated the PDF, formal citations for LLMs (Llama-3.1-8B-Instruct, Mistral-7B-Instruct-v0.2, Qwen2.5-14B-Instruct) are still missing. Providing URLs in footnotes is not a substitute for proper academic citation.
>
> **2. Regarding W2 & Q6:**
> I appreciate the inclusion of Table 6. However, listing empirical improvements is insufficient. The paper lacks an in-depth analysis explaining why these specific loss terms are effective.
>
> **3. Regarding W3 & Q8:**
> I am not convinced by the claim that "adaptation to tasks like summarization or open-ended generation is straightforward," given that no evidence is provided. If the method is truly task-agnostic, demonstrating its effectiveness on a non-QA task with objective ground truth (like Math) would strengthen the claim.
>
> **4. Regarding Q4:**
> The details of the selection head (2-layer MLP) are still missing in the manuscript. This information, along with the specific hidden dimensions used for each model, is crucial for reproducibility. Furthermore, a detailed comparison of computational costs (e.g., training GPU hours and inference latency overhead) should be included rather than just stated as negligible in the rebuttal.
>
> **5. Writting clarity concerns:**
> The term $\ell(\pi_\theta, y_{i,+})$ is used, but the function $\ell$ was previously defined in Line 261-262 as $\ell(\pi_\theta, \pi_{\text{ref}}, y)$. Why is $\pi_{\text{ref}}$ omitted in Eq(4)? Line 340 mentions setting a "confidence threshold of 0.7" for data construction. However, Tables 2 and 3 report different target coverage values ($c$) for different models. Why is the target coverage $c$ different across models in the experiments?
>
> I believe the current version of the paper falls below the acceptance threshold for ICLR. I suggest the authors carefully revise the manuscript.

---

> ### Author Response · Authors · 2025-11-27
>
> We sincerely thank the reviewer for the thoughtful feedback.
>
> **Regarding W1 -- Citations for base LLMs**
>
> A: We have added these citations in the updated PDF.
>
> **Regarding W2 & Q6 -- Why the modified risk terms in Eq.3 are effective:**
>
> A: Traditional DPO depends only on the gap $w(y_+) - w(y_-)$, therefore, the model can minimize the loss by lowering the probability of the chosen answer $\pi_\theta(y_+)$, provided it reduces the probability of the rejected answer $\pi_\theta(y_-)$ even more. This often leads to a phenomenon where the model becomes less likely to generate the correct answer compared to the reference model, effectively "breaking" the model's calibration and utility.
>
> To fix the previous issue, we introduce the additional terms in Equation 3: $l(\pi_\theta, y_{i,+})$ and $l(\pi_\theta, y_{i,-})$. These are losses applied to the absolute values of the rewards, not their difference. Mathematically, these are not the same as DPO. DPO encourages $w(y_+) > w(y_-)$; while SelectLLM’s added terms encourage $w(y_+) \geq 0$ (do not degrade chosen probability below reference) and $w(y_-) \leq 0$ (do not increase rejected probability above reference).
>
> **Regarding W3 & Q8 -- Demonstrating selectLLM's effectiveness on a non-QA task with objective ground truth (like Math)**
>
> A: We thank the reviewer for requesting evidence beyond QA-style benchmarks. To address the reviewer’s suggestion, we evaluated SelectLLM on GSM8K, a high-quality grade-school math problem set. We split the dataset into 6000 training, 1000 validation, and 1000 testing samples, using the same data construction process and training procedure, with the base model Mistral-7B-Instruct-v0.2. The results below demonstrate that SelectLLM continues to improve risk-coverage trade-offs without any task-specific engineering.
>
> | Model        | TP (±)        | TN (±)        | TRUTH (±)     | Precision (±) | Recall (±)   | Coverage (%) (±) |
> |--------------|--------------:|--------------:|--------------:|--------------:|--------------:|------------------:|
> | base | 482.0±3.8    | 0.0±0.0| 482.0±3.8 | 0.482±0.003   | 1.000±0.0   | 100%±0.0 |
> | DPO    | 551.0±12.2    | 0.0±0.0    | 551.0±12.2  | 0.551±0.012 | 1.000±0.0   | 100%±0.0|
> | SelectLLM (c=0.7)    | 558.0±15.6    | 165.0±18.3 | 723.0±16.7 | 0.811±0.012   | 0.922±0.018   | 70.39±1.9 |
>
> As shown, SelectLLM greatly improves precision and overall TRUTH score on Math. This confirms that SelectLLM is task-agnostic and applies directly to non-QA tasks with objectively measurable correctness.
>
> GSM8K: https://huggingface.co/datasets/openai/gsm8k
>
> **Regarding Q4 -- The details of the selection head (2-layer MLP)**
>
> A: Specifically, the selection head consists of:
> 1. Layer-1: A linear projection from the model's hidden size ($d_{model}$) to an intermediate dimension of 512, followed by a ReLU activation and a Dropout layer ($p=0.1$).
> 2. Layer-2: A linear projection from 512 to a scalar output (1), followed by a Sigmoid activation to constrain the confidence score to $(0, 1)$.
>
> Regarding the computational cost, we provide a detailed breakdown of the training and inference costs below using TriviaQA. As shown, the overhead introduced by the selection head is minimal compared to the backbone computation.
>
> | **Metric**                        | **Base Model (DPO Only)** | **SelectLLM** | **Overhead (%)**     |
> |----------------------------------|---------------------------|--------------------------------|-----------------------|
> | **Training Time (GPU Hours)**    | *2.1h*  | *2.15h* | *+2.4%*  |
> | **Inference Latency (per query, TTFT)**| *120ms*          | *120.05ms* | *+0.04%*              |
> | **Parameter Count (Added)**      | 0   | ≈ 2.1M params  | < 0.03% |
>
> 1. Training: Since the gradients for the massive LLM backbone (billions of parameters) dominate the compute, the selection head adds negligible wall-clock time.
>
> 2. Inference: Since the selection head operates on the prefill hidden state of the question, it runs in parallel or immediately preceding the generation. The latency addition is only a single small matrix multiplication.
>
> **Writting clarity concerns -- Eq(4) and the target coverage:**
>
> A: We have updated the Eq(4) in the newest PDF by adding $\pi_\text{ref}$.
>
> Regarding why the coverage values in the tables differ across models, given the fixed 0.7 threshold. It is important to distinguish between the data construction threshold and the target coverage $c$.
> 1. Data Construction Threshold (0.7): As stated in Line 340, this is a fixed threshold used solely to construct the training data. It serves to categorize the answers into "confident" vs. "unconfident". This value is constant for all experiments.
> 2. Target coverage $c$: The parameter $c$ in Equation 5 ($\Psi(c - \hat{\phi}(g))$) represents the training hyperparameter, which is the minimum coverage the model is penalized for failing to reach during optimization. This value is different and tunable for different experiments.

---

> > ### Comment · Reviewer_tA2u · 2025-11-27
> >
> > I thank the authors for their response. But I believe the current quality of the paper falls below the acceptance threshold for ICLR. I suggest the authors carefully revise the manuscript, not just respond to what reviewers suggest.

---

> ### Author Response · Authors · 2025-11-27
>
> We sincerely thank the reviewer for their continued engagement and for holding our work to a high standard. We have carefully revised the manuscript to address every point raised in your previous review. We believe the current version is significantly stronger, with added analysis, architectural details, and new experiments on the mathematical benchmark.
>
> Below, we detail exactly where they can be found in the updated revision:
>
> 1. **Formal Citations (Addressing W1):** We replaced footnotes with formal citations for Llama-3.1, Mistral-v0.2, and Qwen2.5 in the main text (Line 326, 327, Section 5.1) and References.
> 2. **Analysis of Loss Terms (Addressing W2 & Q6)** We added an analysis around Lines 276–283 (Section 4.2), explaining that our risk terms are effective. Appendix B (Table 7) provides the ablation study confirming these terms are critical for maintaining Recall.
> 3. **Generalization: New GSM8K Experiments (Addressing W3 & Q8)** To prove task-agnosticism, we added Section 5.4 (Lines 479–485) and Table 6 evaluating GSM8K (Math). SelectLLM achieves 723.0 TRUTH (vs. 551.0 for DPO), demonstrating robustness on math tasks with verifiable ground truth.
> 4. **Architecture & Cost Details (Addressing Q4)** We added Section 4.1 (Lines 227–235) specifying the selection head as a 512-dimensional MLP. We also added a cost analysis (Lines 236–242), clarifying that the head adds negligible parameters and zero latency overhead during inference.
> 5. **Notation & Threshold Clarity (Addressing Writing Concerns)** Notation: We corrected Equation 4 (Line 289) to explicitly include the input $\pi_\text{ref}$. Thresholds: We clarified that the "0.7 threshold" (Line 359) applies only for data labeling. The target coverage $c$ (Eq. 5) is a user-defined hyperparameter that varies to match the intrinsic difficulty of each dataset (Lines 300).

---

### Official Review · Reviewer_DpC2 · 2025-10-27

**Soundness:** 1
**Presentation:** 1
**Contribution:** 1
**Rating:** 2
**Confidence:** 4

**Summary:**

This paper proposes a confidence-head training strategy combined with the DPO loss. Their method is training a different head, which is for predicting the confidence and DPO loss for the generation. They evaluate their method on various datasets.

**Strengths:**

- Confidence Estimation is a timely and important topic.

**Weaknesses:**

- The paper is poorly written: The novelty is not properly presented. Some technical terms are misused, such as reward function at line 259. Sections are not separated properly: Section 4 only has one subsection, so why do you have a subsection?
- In line 256, the proposed loss addition to DPO is exactly the same loss in DPO because. logx-logy = log(x/y).
- Why do you use an additional model to set the ground truth confidence?
 - If the abstention decision is solely based on the confidence score coming from the confidence head, why do you combine it with DPO preference tuning?
- What is the purpose of defining the expected coverage? Do you evaluate it on your experiments?
- Please report TP/TN ratio rather than the actual magnitude.
- What is the novelty/new perspective of this paper with respect to the current literature?
- Citation of MARS on line 308 is wrong.

**Questions:**

See weaknesses.

---

> ### Author Response · Authors · 2025-11-25
> **Part 1/N**
>
> We thank the reviewer for the review. Here are our responses:
>
> **Q1: The paper is poorly written: The novelty is not properly presented. Some technical terms are misused, such as reward function at line 259. Sections are not separated properly: Section 4 only has one subsection, so why do you have a subsection?**
>
> A1: We understand that the reward is mostly used in the context of reinforcement learning, and it may be misleading in this context. Therefore, we renamed the “reward function” to the “margin function” in our updated PDF.
>
> **Q2: In line 256, the proposed loss addition to DPO is exactly the same loss in DPO because. logx-logy = log(x/y).**
>
> A2: We respectfully point out a fundamental misunderstanding regarding the mathematical formulation of our proposed loss compared to the standard DPO loss.
>
> While the reviewer is correct that the underlying term $w(y) = \beta \log \frac{\pi_\theta(y)}{\pi_{ref}(y)}$ is central to DPO, the loss functions applied to this term in SelectLLM are distinct and serve different optimization objectives.
>
> Because DPO depends only on the gap $w(y_+) - w(y_-)$, the model can minimize the loss by lowering the probability of the chosen answer $\pi_\theta(y_+)$, provided it reduces the probability of the rejected answer $\pi_\theta(y_-)$ even more. This often leads to a phenomenon where the model becomes less likely to generate the correct answer compared to the reference model, effectively "breaking" the model's calibration and utility.
>
> To fix the previous issue, we introduce the additional terms in Equation 3: $l(\pi_\theta, y_{i,+})$ and $l(\pi_\theta, y_{i,-})$. These are losses applied to the absolute values of the rewards, not their difference. Mathematically, these are not the same as DPO. DPO encourages $w(y_+) > w(y_-)$; while SelectLLM’s added terms encourage $w(y_+) \geq 0$ (do not degrade chosen probability below reference) and $w(y_-) \leq 0$ (do not increase rejected probability above reference).
>
> **Q3: Why do you use an additional model to set the ground truth confidence?**
>
> A3: We clarify that we do not use the external model to provide a ground-truth confidence target for regression for the confidence head, nor do we distill its probability distribution. Instead, we use the external model (DeepSeek-v3) solely as a data filter during the dataset construction phase. We apply a threshold to the "tone-confidence" from the external model to categorize answers into binary 'chosen' and 'rejected' pairs (Section 5.1) for DPO fine-tuning.
>
> After the filtering, the confidence scores from the additional model are discarded completely. The confidence score produced by SelectLLM is not trained to mimic the external model's score. Instead, it is learned end-to-end via our selection head $g(\cdot)$. This head optimizes the risk-coverage trade-off defined in Equation 5, learning to assign high scores to questions where the model itself is likely to produce a correct answer.
>
> **Q4: If the abstention decision is solely based on the confidence score coming from the confidence head, why do you combine it with DPO preference tuning?**
>
> A4: As emphasized in our title and Introduction, the central premise of SelectLLM is the joint optimization of the risk-coverage trade-off. Treating the confidence estimation and the generation quality as separate problems ignores the fact that a model’s utility depends on both its ability to generate correct answers and its ability to identify them.
>
> In the original paper, we have explained the necessity of this combination in the last paragraph of Section 4. Specifically, without the $L_{DPO}$ loss forcing the model to learn from all training instances, the selection head tends to "focus on a fraction $c$ of the training set before accurate low-level features are constructed", causing it to overfit to the wrong subset of data. The DPO loss acts as an essential regularizer, exposing the model to the full dataset to ensure robust feature learning. Joint training allows the model to simultaneously improve its answer quality (increasing potential coverage) and its self-evaluation (reducing risk).

---

> ### Author Response · Authors · 2025-11-25
> **Part 2/2**
>
> **Q5: What is the purpose of defining the expected coverage? Do you evaluate it on your experiments?**
>
> A5: The "expected coverage" parameter $c$ serves a critical purpose in our loss function (Equation 5 in the updated PDF): it acts as a constraint to prevent the trivial solution where the model achieves zero risk by simply abstaining from every question.
> Following the established framework of SelectiveNet (https://arxiv.org/pdf/1901.09192), which we cite in our paper, the objective of selective prediction is to minimize risk subject to a coverage constraint. Without the term $\Psi(c - \hat{\phi}(g))$ penalizing the model for falling below target coverage $c$, the selection head would converge to outputting zero confidence for all inputs to minimize the selective risk to zero.
>
> To evaluate the impact of the target coverage rate $c$ on model performance, we plot a risk-coverage curve with target coverage rates $c \in \\{0.7, 0.75, 0.8, 0.85, 0.9, 0.95\\}$ using Llama-3.1-8B-Instruct on the TriviaQA dataset. This figure (Figure 4(c)) is included in the Appendix of our revised PDF. The results demonstrate that while SelectLLM achieves the highest TRUTH score (TP + TN) at $c=0.85$ (the setting used in our main experiments), its performance remains robust across other values. Crucially, SelectLLM consistently and significantly outperforms the DPO baseline across all tested coverage rates.
>
> **Q6: Please report TP/TN ratio rather than the actual magnitude.**
>
> A6: Since our test set consists of exactly 1,000 samples, the reported absolute values for TP and TN are directly interpretable as percentages (normalized by the total number of samples). For example, the results of SelectLLM with llama3.1 and TriviaQA are -- TP: 58.2%; TN: 17.0%; TRUTH: 75.2%.
>
> **Q7: What is the novelty/new perspective of this paper with respect to the current literature?**
>
> A7: SelectLLM is the first method to jointly optimize an LLM for the risk-coverage trade-off. Towards this goal, SelectLLM employs a novel combination of DPO with explicit confidence estimation.
>
> We summarize our specific contributions and novelty relative to the literature as follows:
> 1. Existing alignment methods (e.g., standard DPO, PPO) optimize solely for generation quality (utility). Existing uncertainty methods (e.g., Semantic Entropy, P(True)) typically operate post-hoc on frozen models. SelectLLM bridges this gap by introducing a joint objective (Equation 6) that simultaneously fine-tunes the model to generate better answers and trains a dedicated head to recognize when it cannot. This explicitly optimizes the Risk-Coverage trade-off during training, which previous methods do not address directly.
> 2. We introduce a novel dual-head architecture (decoding head + selection head) tailored for sequences. Crucially, we modify the loss landscape by adding absolute risk constraints (Equation 4) to the standard relative DPO loss. This prevents the "likelihood displacement" issue where standard DPO might degrade the absolute probability of correct answers, ensuring the selection head receives stable signals.
> 3. Unlike sampling-based baselines or external-model methods, SelectLLM is self-contained. It computes a reliable confidence score in a single forward pass without external dependencies. Furthermore, our experiments demonstrate a novel capability: the learned abstention mechanism generalizes to out-of-distribution (OOD) domains (e.g., MedConceptsQA).
>
> **Q8: Citation of MARS on line 308 is wrong**
>
> A8: We thank the reviewer for pointing this out. We have updated the citation of MARS in our updated PDF.

---

> > ### Author Response · Authors · 2025-11-27
> > **Kind Follow-up on Our Rebuttal**
> >
> > Dear reviewer DpC2,
> >
> > Thank you for taking the time to review our work. As a gentle reminder, the discussion phase concludes on December 2. We would be more than happy to provide further clarification if you have any additional questions. Thank you again!
> >
> > Authors of SelectLLM

---

### Official Review · Reviewer_KetN · 2025-10-27

**Soundness:** 2
**Presentation:** 1
**Contribution:** 1
**Rating:** 2
**Confidence:** 3

**Summary:**

This paper proposes a method for training a confidence-estimation head in addition to the standard autoregressive decoding loss for an LLM. The paper proposes to use this confidence-estimation head to gate whether the model abstains from providing an answer. The loss function that is used to train the model is a combination of a DPO loss and a ‘select’ loss that attempts to maintain a target global coverage (i.e. abstention vs non-abstention) rate. The authors show that their approach achieves higher performance (defined by the sum of the rate of true positives and true negatives) than a range of prior methods, on three Q&A datasets, and across three open-source models.

Overall, in my view, there are several missing details in this paper, that make it difficult to properly evaluate. I hope that the authors are able to provide some of these details, as well as comprehensively answer my set of questions in the weaknesses below.

**Strengths:**

1. Strong performance on relevant metrics (TRUTH score).
2. Extensive set of baselines that are compared to.

**Weaknesses:**

There are several weaknesses in the paper, some of which I think are quite serious:

1. Line 270 states that “In the appendix, we include an ablation study to demonstrate the effectiveness of the two additional terms”, but there is no such appendix included.
2. There are missing experimental details, including hyperparameters such as learning rate, training epochs, optimizer. Most crucial is the missing batch size, because:
3. It is not clear how the empirical coverage (line 277) is calculated. Presumably this is not recomputed on the entire dataset at every training iteration. If it is estimated by the empirical coverage within a batch, then it is crucial to have a high enough batch size for this to be a low variance estimator; and I would like to have seen an ablation w.r.t. batch size.
4. What is the target coverage rate that is used in the experiments? I did not see this detailed anywhere.
5. It is also not clear to me exactly how the dataset is constructed. What constitutes a preferred and a dispreferred response? The statement is that a threshold of 0.7 is used and any response assigned a confidence score above that is considered ‘accepted’ and those below ‘rejected’. In which case, how are the pairs of {preferred, dispreferred} constructed precisely? If the pairing does not matter (e.g. it is done at random), then the use of DPO is not the most suitable choice to use; general margin maximisation algorithms such as [KTO](https://arxiv.org/abs/2402.01306) may be more appropriate.
6. The paper makes extensive reference to ‘human preferences’, yet, the experiments use a strong LLM (DeepSeek v3) to mark the confidence of the responses. The motivation of this design choice is not discussed in the paper at all. The use of this strong LLM labeller suggests that the method primarily benefits from distillation of confidence from a much stronger LLM.
7. Related to point 6) above, a standard approach of fine-tuning with LoRA to calibrate the LLM directly (as done in e.g. [[1]](https://arxiv.org/abs/2207.05221), [[2]](https://proceedings.neurips.cc/paper_files/paper/2024/file/9c20f16b05f5e5e70fa07e2a4364b80e-Paper-Conference.pdf) should be compared to the method used. It is claimed that the latter is done but there are no details given on the methodology used there, so it is impossible to judge if it is a fair comparison to the proposed method.
8. The discussion of and adjustment to the DPO loss function due to the possibility of the preferred responses’ likelihoods being reduced by the standard loss function would benefit from reference to prior works such as [[3]](https://arxiv.org/abs/2404.12358), [[4]](https://arxiv.org/abs/2404.04626), [[5]](https://arxiv.org/pdf/2402.13228), which all identify the issue as well as provide suggested ameliorations.

**Questions:**

See weaknesses above.

---

> ### Author Response · Authors · 2025-11-25
> **Part 1/N**
>
> We thank the reviewer for the review. Here are our responses:
>
> **Q1: Line 270 states that “In the appendix, we include an ablation study to demonstrate the effectiveness of the two additional terms”, but there is no such appendix included.**
>
> A1: We thank the reviewer for pointing this out. We have included the missing ablation study in Table 6 in the updated PDF.
>
> **Q2 & Q3: There are missing experimental details, including hyperparameters such as learning rate, training epochs, optimizer. Most crucial is the missing batch size, because: It is not clear how the empirical coverage (line 277) is calculated. Presumably this is not recomputed on the entire dataset at every training iteration. If it is estimated by the empirical coverage within a batch, then it is crucial to have a high enough batch size for this to be a low variance estimator; and I would like to have seen an ablation w.r.t. batch size.**
>
> A2:  To evaluate the impact of the learning rate $lr$ on model performance, we plot a risk-coverage curve with target coverage rates $lr \in \\{1e-7, 5e-7, 1e-6, 2e-6, 5e-6\\}$ using Llama-3.1-8B-Instruct on the TriviaQA dataset. This figure (Figure 4(d)) is included in the Appendix of our revised PDF. The results demonstrate that while SelectLLM achieves the highest TRUTH score (TP + TN) at $lr=1e-6$ (the setting used in our main experiments), its performance remains robust across other values. Crucially, SelectLLM consistently and significantly outperforms the DPO baseline across all tested coverage rates.
>
> The reviewer is correct that the empirical coverage $\hat{\phi}(g)$ is calculated as the average coverage within the current mini-batch, not the entire dataset, at each iteration.
>
> To evaluate the impact of the batch size $bsz$ on model performance, we plot a risk-coverage curve with target coverage rates $bsz \in \\{30, 60, 120, 180, 240\\}$ using Llama-3.1-8B-Instruct on the TriviaQA dataset. This figure (Figure 4(e)) is included in the Appendix of our revised PDF. The results demonstrate that SelectLLM achieves the highest TRUTH score (TP + TN) at $bsz=240$ (due to the limited GPU resources, the batch size used in our main experiments is 120). Additionally, although a larger batch size can achieve a higher TRUTH score, a smaller batch size (e.g., 30) reduces it by at most 4%, showing that the batch-level estimation of coverage did not introduce significant destabilizing noise into the training process.
>
> Regarding the training epoch and optimizer, we empirically observed that extending training beyond 1 epoch resulted in significant performance degradation. This is consistent with findings in prior preference alignment literature (e.g., DPO), where multi-epoch training often leads to overfitting on the small preference dataset and a degradation of the model's general capabilities. We utilized paged_adamw_32bit primarily to optimize memory usage during QLoRA fine-tuning. We also conducted validation tests using standard adamw and observed comparable convergence behaviour and final performance, confirming that the specific choice of optimizer variant did not introduce bias into our results.
>
> **Q4: What is the target coverage rate that is used in the experiments? I did not see this detailed anywhere.**
>
> A3: We have added the target coverage rate for each base model and task to the Tables in the updated PDF.
>
> To further evaluate the impact of the target coverage rate $c$ on model performance, we plot a risk-coverage curve with different target coverage rates $c \in \\{0.85, 0.8, 0.75, 0.7, 0.65\\}$ using Llama-3.1-8B-Instruct on the TriviaQA dataset. This figure is included in the Appendix of our revised PDF. The results demonstrate that while SelectLLM achieves the highest TRUTH score (TP + TN) at $c=0.75$ (the setting used in our main experiments), its performance remains robust across other values. Crucially, SelectLLM consistently and significantly outperforms the DPO baseline across all tested coverage rates.

---

> > ### Author Response · Authors · 2025-11-25
> > **Part 2/N**
> >
> > **Q5: It is also not clear to me exactly how the dataset is constructed. What constitutes a preferred and a dispreferred response? The statement is that a threshold of 0.7 is used and any response assigned a confidence score above that is considered ‘accepted’ and those below ‘rejected’. In which case, how are the pairs of {preferred, dispreferred} constructed precisely? If the pairing does not matter (e.g. it is done at random), then the use of DPO is not the most suitable choice to use; general margin maximisation algorithms such as KTO may be more appropriate.**
> >
> > A4: We clarify the exact construction of the preference pairs and justify the use of DPO over KTO.
> >
> > The pairing is not random; it is strictly conditional on the specific input question $x$. As described in Section 5.1, the construction process for each question is as follows:
> > 1. Candidate Generation: For a given question $x$, we generate multiple candidate answers $\{\hat{y}_1, \hat{y}_2, ...\}$.
> > 2. Scoring & Labeling: We assign a confidence score to each answer using the external model (tone-confidence) and verify correctness against the ground truth.
> > Chosen Set ($y_+$): Answers that are both factually correct AND have a confidence score >= 0.7.
> > Rejected Set ($y_-$): Answers that are incorrect OR have a confidence score <0.7.
> > 3. Fallback Mechanism: If the model cannot produce any correct, high-confidence answer for question $x$, the "Chosen" response ($y_+$) defaults to the refusal string: "I don't know the answer". This ensures the model prefers abstention over hallucination.
> > 4. Pairing: We construct the training instance $(x, y_+, y_-)$ by sampling one response from the Chosen set and one from the Rejected set for the same question $x$
> >
> > KTO (Kahneman-Tversky Optimization) is typically advantageous when preference data is unpaired (i.e., we only know if a response is "good" or "bad" in isolation, without a direct comparison for the same prompt). In our case, we explicitly construct paired data: we have two competing answers ($y_+$ and $y_-$) for the exact same input context $x$. The objective is to teach the model to distinguish between a good response (correct/confident or "I don't know") and a bad one (hallucinated or under-confident) given that specific query. DPO is designed precisely to optimize this relative margin, making it the most suitable algorithmic choice for our paired setup.
> >
> >
> > **Q6: The paper makes extensive reference to ‘human preferences’, yet, the experiments use a strong LLM (DeepSeek v3) to mark the confidence of the responses. The motivation of this design choice is not discussed in the paper at all. The use of this strong LLM labeller suggests that the method primarily benefits from distillation of confidence from a much stronger LLM.**
> >
> > A5: We reference "human preferences" to describe the alignment objective. Specifically, the preference for trustworthy, calibrated responses over hallucinated ones. We employ an LLM as the annotator purely for scalability. Obtaining human annotations for millions of training pairs to distinguish subtle "tone" differences is prohibitively expensive. An automated annotator allows us to scale the supervision necessary for the model to learn generalization.
> >
> > In terms of the "Distillation" hypothesis, we respectfully disagree that our method relies on distilling the knowledge of a "stronger" LLM. Two key factors refute this:
> >
> > 1. The Teacher Does Not Provide "Correctness": DeepSeek-v3 is not used to determine if an answer is correct. It is strictly used to classify the tone (confident vs. unconfident). The actual signal for "correctness", which is the hardest part to learn, comes from the dataset's ground truth labels (the gold answers in TriviaQA), not from DeepSeek. SelectLLM learns to align its internal confidence with this ground truth, not with DeepSeek’s knowledge.
> >
> > 2. Independence from Teacher Strength:
> > To empirically validate the robustness of SelectLLM to the choice of the annotator model, we conducted an additional experiment using Qwen3-32B to reconstruct the TriviaQA preference dataset. Qwen3-32B is significantly smaller and less powerful than the DeepSeek-v3 model used in our main experiments. We then retrained the Llama-3.1-8B-Instruct model using SelectLLM on this new dataset.
> >
> > | Model        | TP (±)        | TN (±)        | TRUTH (±)     | Precision (±) | Recall (±)   | Coverage (%) (±) |
> > |--------------|--------------:|--------------:|--------------:|--------------:|--------------:|------------------:|
> > | DeepSeek-v3  | 582.0±19.7    | 170.0±25.2    | 752.0±2.6     | 0.773±0.015   | 0.884±0.021   | 75.96±3.63        |
> > | Qwen3-32B    | 570.0±12.6    | 183.0±19.4    | 753.0±1.7     | 0.762±0.012   | 0.902±0.018   | 75.33±2.78        |
> >
> > This empirically demonstrates that the method does not rely on the "intelligence" or "strength" of the labeller, but rather on the structure of the training data (pairing correct/confident answers against incorrect ones).

---

> ### Author Response · Authors · 2025-11-25
> **Part 3/3**
>
> **Q7: Related to point 6) above, a standard approach of fine-tuning with LoRA to calibrate the LLM directly (as done in e.g. [1], [2] should be compared to the method used. It is claimed that the latter is done but there are no details given on the methodology used there, so it is impossible to judge if it is a fair comparison to the proposed method.**
>
> A6: We appreciate the opportunity to clarify the implementation details of our fine-tuning baseline to demonstrate the fairness of the comparison.
>
> The "standard approach of fine-tuning to calibrate" corresponds to the LACIE baseline in our experiments. This method uses DPO to directly fine-tune the LLM to output refusal strings (e.g., "I don't know") when the answer is incorrect or uncertain, effectively "calibrating" the model's generation behaviour.
>
> To ensure a strictly fair, "apples-to-apples" comparison, we implemented LACIE using the identical experimental setup as SelectLLM. Specifically: LACIE was fine-tuned on the exact same set of preference pairs $\{(x, y_+, y_-)\}$ described in Section 5.1. SelectLLM uses these pairs to train the selection head and the backbone. LACIE uses these pairs to train the backbone to prefer the "Chosen" response (which is a refusal string if the model is uncertain) over the "Rejected" response. Additionally, both models utilize QLoRA with the same rank ($r=16$) settings.
>
> Comparison: Since the training data, backbone, and parameter budget were controlled, the significant performance gap (e.g., TriviaQA TRUTH: 752.0 for SelectLLM vs. 579.3 for LACIE, Table 2) is directly attributable to the methodology. LACIE relies on the model's ability to "verbalize" uncertainty via next-token prediction, which often conflicts with the objective of generating correct answers. In contrast, SelectLLM's dedicated selection head provides a continuous, decoupled signal that enables more precise and flexible abstention without degrading generation quality.
>
> **Q8: The discussion of and adjustment to the DPO loss function due to the possibility of the preferred responses’ likelihoods being reduced by the standard loss function would benefit from reference to prior works such as [3], [4], [5], which all identify the issue as well as provide suggested ameliorations.**
>
> A7: We thank the reviewer for highlighting these relevant prior works. We have updated the Method (Section 4.1) in the revised PDF to explicitly cite these references.

---

> > ### Author Response · Authors · 2025-11-27
> > **Kind Follow-up on Our Rebuttal**
> >
> > Dear reviewer KetN,
> >
> > Thank you for taking the time to review our work. As a gentle reminder, the discussion phase concludes on December 2. We would be more than happy to provide further clarification if you have any additional questions. Thank you again!
> >
> > Authors of SelectLLM

---

### Official Review · Reviewer_atUF · 2025-11-01

**Soundness:** 2
**Presentation:** 2
**Contribution:** 3
**Rating:** 8
**Confidence:** 4

**Summary:**

The paper introduces SelectLLM, an end-to-end method to calibrate LLMs for selective prediction by adding a selection head and training to balance coverage and risk, claiming strong results on TriviaQA, CommonsenseQA, MedConceptsQA.

It begins with framing the risk–coverage trade-off in terms of four different outcome types (accept , reject) x  (correct, incorrect). It then motivates the need and pitfalls of abstention, and positions the proposed methods as optimizing that trade-off. The high level idea is to attach a selection head that reads the last hidden state of the question to produce a question-level confidence. The decoding head is trained with DPO, the selection head is optimized for risk–coverage, some illustrative cases are shown in table 1. Moving to more details into the method, the idea is to formalize coverage as proportion answered and risk as error rate over answered set, and then suggests DPO training with pairwise preferences, with the goal of a target coverage x and to abstain if the confidence is below a threshold. The selection head outputs confidence (between 0 and 1) from the last question token state. The loss is a combination of a DPO loss and a select loss. A modified empirical selective risk is defined with some additional terms (line 271), using a reward that prevents DPO from decreasing both probabilities. The two losses are combined with a detault weight of 0.5 (i.e. takes a simple average).

For experiments, the following base models are chosen: Llama-3.1-8B-Instruct, Mistral-7B-Instruct-v0.2, Qwen2.5-14B-Instruct; QLoRA rank-16. The proposed method is compared against base, LACIE (DPO), LARS, MARS, TokenSAR, P(True), Semantic Entropy. The metrics used are TP/TN/Precision/Recall/Coverage, and TRUTH = TP+TN (upper bound 1000). For score-based methods, tune the threshold on validation to maximize TRUTH, then apply to test; non-score baselines accept anything unless model explicitly refuses. It is reported that SelectLLM substantially increases TN and Precision at the cost of Coverage/Recall; TRUTH is best across models/datasets. Some experiments for OOD are also reported.

**Strengths:**

The paper has a clear practical goal and contribution -- that of using train-time selective prediction for LLMs in a principled way, not just test-time thresholding.

The architecture proposed is simple and general. It has a single selection head with minimal changes needed to the decoding pipeline.

The results show consistent empirical gains in TRUTH and precision across three base models and two ID datasets. The OOD results are also encouraging.

**Weaknesses:**

This is not really a weakness but cross my mind. Confidence uses only the question’s last token hidden state, it ignores evidence in the generated answer or early decoding signals. These can be highly informative for difficulty/uncertainty. While I understand that the authors made a choice, it may systematically miss cases where uncertainty emerges during generation (such as in multi-hop). Further, the authors argue that token probs are miscalibrated so a separate head is needed. However, then that head is trained without using token-level uncertainty or decoding statistics. I wonder if calibration performance could be better if so.

Slight consistency issue: Earlier in the paper coverage is defined as the fraction answered. However, later, the empirical coverage is defined as the average of g(h), where g outputs a confidence score between 0 and 1 (not a binary accept indicator). This makes the empirical coverage an average score, not the fraction answered. This causes a slight consistency mismatch in what is argued and e.g. in the constraint terms (equation 4).

There seems to be a circularity or leakage risk in the validation of confidence. Training pairs are constructed using DeepSeek-v3 "tone-confidence" thresholds and fallbacks. But later, SelectLLM’s confidence is validated by comparing to the same tone-confidence distribution (Fig. 3). This is not an independent validation signal. The risk is that it could be teaching the model to mimic tone signals and then "validating" that mimicry. For such empirical work this is a chicken and egg problem, but it would at least be useful to spell out.

Using LDPO (pairwise margin) inside the selective risk is an indirect surrogate for correctness and might misalign with the risk–coverage target, especially OOD. (pages 5–6)

**Questions:**

See above.

In addition.

Why restrict g() to the question state only? Did you test variants that pool answer states or incorporate token-entropy/semantic entropy/variance across samples? If tested, please share,

---

> ### Author Response · Authors · 2025-11-25
> **Part 1/N**
>
> We thank the reviewer for the review. Here are our responses:
>
> **Q1: Why does SelectLLM rely solely on the question(last-token) hidden state and not generation-time features?**
>
> A1: We agree that including generated answer tokens or decoding statistics could provide additional signals for difficulty. But our goal is to allow the model to decide whether it should answer a question before generating an answer. Using decoding-time signals would force the model to partially generate a response before abstaining, which is undesirable in safety-critical or latency-sensitive applications (e.g., medical decision support). This motivates a question-level reject option rather than a generation-level reject. Additionally, we aim to capture the model's intrinsic "self-knowledge" regarding the prompt, i.e., does the model know this concept, rather than its confidence in a specific sampled generation path.
>
> Regarding the multi-hop uncertainty, we agree that some uncertainty emerges during generation. However, using answer-side signals can artificially inflate coverage, because the model can “cheat” by using partial answer attempts to score confidence. This undermines the purpose of abstention.
>
> In terms of incorporating token probabilities if they contain uncertainty, we explicitly avoid them because they are miscalibrated. Using them as features would leak the miscalibration back into the selection head, precisely what we aim to correct. Instead, we learn a separate predictor that is independent from token-level biases, making the selection head act as a calibration layer rather than inheriting uncertain logits.
>
> Our current work establishes the value of question-only selective prediction, and integrating decoding statistics is a promising next step.
>
> **Q2: Consistency of coverage vs. empirical coverage**
>
> A2: We appreciate the reviewer pointing out this potential confusion. This is a standard relaxation technique used in selective prediction literature to make the objective differentiable.
>
> Theoretical Definition: Coverage is indeed the fraction of answered questions (Binary: $\frac{1}{n} \sum \mathbb{I}(g(h_i) > \tau)$).
>
> Training Definition: To optimize this via gradient descent, we use the "soft" empirical coverage, defined as the average of the continuous confidence scores: $\hat{\phi}(g) = \frac{1}{n} \sum g(h_i)$.
> This acts as a differentiable proxy for the binary count.
>
> This matches prior selective-classification work (e.g., SelectiveNet), which also optimizes soft coverage and thresholds at test time.
>
> **Q3: Circularity” concern regarding validation using tone-confidence**
>
> A3: We respectfully clarify that our method does not rely on “mimicry” of DeepSeek-v3 scores, nor does it regress toward external confidence labels. DeepSeek-v3 signals are used only to derive preference pairs for DPO fine-tuning, analogous to how prior work uses human preference labels. SelectLLM is not trained to predict confidence bins, and no loss term uses tone values directly.
>
> During evaluation, we do not compare absolute scores; instead, we plot distributions to show that the model has learned a meaningful ordering of uncertainty without access to external scores. In both training and validation, ground-truth correctness remains available from the base dataset, and we use disjoint splits, following standard experimental practice. Thus, the validation procedure does not leak supervision and does not induce circular reinforcement of DeepSeek-v3 predictions.

---

> > ### Author Response · Authors · 2025-11-25
> > **Part 2/2**
> >
> > **Q4: Selective risk uses $L_{DPO}$: is it an indirect surrogate?**
> >
> > A4: We agree that $L_{DPO}$ is an indirect surrogate for correctness, but this is intentional and principled. SelectLLM must jointly optimize (a) when to answer and (b) how well it answers. Using explicit correctness supervision would require labeled ground-truth answers, whereas our objective is to leverage preference-based alignment signals, a key advantage of DPO-style training.
> >
> > Using $L_{DPO}$  in the selective-risk objective encourages the selection head to abstain precisely when the model is likely to violate the preference objective. While surrogate objectives can, in principle, misalign under distribution shifts, our empirical results on MedConceptsQA show the opposite: the abstention behavior transfers robustly to unseen domains, indicating that the learned mapping “low loss <-> high confidence&accuracy” generalizes beyond the training distribution.
> >
> > We would welcome clarification from the reviewer on what specific mechanism they believe could cause a surrogate objective like $L_{DPO}$ to misalign under OOD conditions, as such insight would help us further strengthen the work.
> >
> > **Q5: Why restrict g() to the question state only? Did you test variants that pool answer states or incorporate token-entropy/semantic entropy/variance across samples? If tested, please share**
> >
> > A5: We did not explicitly integrate answer-pooling or entropy features into the SelectLLM architecture because doing so would fundamentally compromise one of the method's primary contributions: computational efficiency via pre-generation abstention. This allows the system to save significant compute by "early exiting" on questions it knows it cannot answer. Incorporating answer states, token entropy, or variance (like Semantic Entropy) requires generating the full response (or multiple responses) first, effectively doubling or multiplying the inference cost. But we agree it is an interesting direction for future work.

---

> > > ### Comment · Reviewer_atUF · 2025-11-25
> > > **Ack**
> > >
> > > Thanks for the detailed responses. These have helped clarify some of my confusions. I will keep my rating (at a clear accept). I think the proposed method is useful.

---

> > > > ### Author Response · Authors · 2025-11-25
> > > >
> > > > Thank you very much for taking the time to review our work and for maintaining the positive score. We appreciate your careful consideration.

---

### Author Response · Authors · 2025-11-29
**General Response (1/2)**

Below, we provide a consolidated meta-rebuttal addressing all questions across four reviews, grouped by five themes. In response to the reviewer criticisms, we have
1. Clarified architectural details and formal definitions,
2. Performed detailed ablation studies on batch size, learning rates, and loss terms,
3. Conducted a robustness study using a smaller annotator (Qwen3-32B) to disprove "distillation" concerns, and
4. Conducted new experiments on GSM8K to demonstrate task-agnostic generalization.


We have revised our manuscript accordingly, highlighting the major modifications, and uploaded it on OpenReview.

In conclusion, we believe that we have addressed all major criticisms of the reviewers and kindly request that our score be increased.

We organize our responses by key themes as follows:

**Theme 1: Methodological Novelty & Loss Formulation**
Questions addressed:
*R DpC2 Q2:* Claims the proposed loss addition is mathematically identical to standard DPO.
*R atUF Q4:* Is using $L_{DPO}$ in selective risk an indirect surrogate?
*R DpC2 Q4:* Why combine the selection head with DPO preference tuning?
*R DpC2 Q7:* What is the novelty relative to current literature?
*R tA2u Q7:* Which component is most critical (head, coverage constraint, or modified risk)?

**Response:**
There is a fundamental difference between SelectLLM and standard DPO. Standard DPO depends only on the relative gap $w(y_+) - w(y_-)$. This allows the model to minimize loss by lowering the probability of the chosen answer, provided it lowers the rejected answer more. This "likelihood displacement" breaks calibration.

To fix this, SelectLLM introduces absolute risk constraints (Eq. 3). We apply losses to the absolute values of the rewards: $l(\pi_\theta, y_{i,+})$ encourages $w(y_+) \geq 0$ (do not degrade chosen probability) and $l(\pi_\theta, y_{i,-})$ encourages $w(y_-) \leq 0$. This is not mathematically identical to DPO. Our ablation study also shows its effectiveness.

We combine this with a separate Selection Head to jointly optimize the Risk-Coverage trade-off (Eq. 6). Without the joint $L_{DPO}$ loss, the selection head overfits to a fraction of the data; without the coverage constraint, the head collapses to zero coverage. Our ablation studies confirm that removing the modified risk terms or the selection head significantly degrades performance (e.g., TriviaQA TRUTH score drops from 752.0 to 579.3 when removing the head, i.e., the LACIE baseline).

In conclusion, we argue that the novelty and soundness of SelectLLM is greater than the scores by reviewers DpC2 and atUF.

**Theme 2: Dataset Construction, "Distillation," and Annotator Robustness**
Questions addressed:
*R atUF Q3:* "Circularity" concern regarding validation using tone-confidence.
*R KetN Q6:* Concern that the method relies on distillation from a strong LLM (DeepSeek).
*R tA2u Q1:* How is performance affected if a less powerful LLM is used for labeling?
*R tA2u Q2:* Reliability of tone-confidence labels.
*R KetN Q5:* How are preference pairs constructed?
*R DpC2 Q3:* Why use an additional model for ground truth confidence?

**Response:**
We emphasize that our method does not rely on distilling the intelligence or probability distributions of the stronger external model (DeepSeek-v3). As detailed in Section 5.1, the construction pipeline proceeds as follows:

1. For a given question $x$, we first generate multiple candidate answers $\{\hat{y}_1, \hat{y}_2, ...\}$.

2. We then assign a (tone-)confidence score to each answer using the external model (DeepSeek-v3) and verify correctness against the ground truth from the original dataset. For DPO, the chosen set ($y_+$) consists of answers that are both factually correct AND have a (tone-)confidence score >= 0.7; whereas the rejected set ($y_-$) contains answers that are incorrect OR have a (tone-)confidence score <0.7.

Therefore, the external model is used solely as a tone classifier during dataset construction. It only classifies the answer's "tone" (confident vs. unconfident), not its factual correctness. Factual correctness is determined strictly by the independent, external ground-truth labels, which are external to both the answer-generating model and the tone-classifier model.

To empirically validate the robustness of SelectLLM to the choice of the external model, we constructed another version of the TriviaQA dataset using a significantly smaller model, Qwen3-32B, as the (tone-)confidence scorer. The performance remained virtually identical:

| **Annotator Model** | **TRUTH Score**     | **Coverage (%)**     |
|---------------------|----------------------|------------------------|
| DeepSeek-v3         | 752.0 ± 2.6          | 75.96 ± 3.63           |
| Qwen3-32B           | 753.0 ± 1.7          | 75.33 ± 2.78           |

This confirms that SelectLLM relies on the training objective, not the additional model’s strength.

---

> ### Author Response · Authors · 2025-11-29
> **General Response (2/2)**
>
> **Theme 3: Generalization (New Math Experiments) & Architecture**
> Questions addressed:
> *R tA2u W3 & Q8:* Does the method generalize to non-QA tasks like Math?
> *R atUF Q1 & Q5:* Why rely on the question-only state? Why not pool answer states/entropy?
> *R tA2u Q4:* Architecture of the selection head and inference cost.
>
> **Response:**
> Generalization: We evaluated SelectLLM on GSM8K (Grade School Math) to demonstrate task-agnosticism. SelectLLM significantly outperforms the DPO baseline on reasoning tasks:
> *DPO Baseline:* TRUTH Score 551.0
> *SelectLLM:* TRUTH Score 723.0
>
> Architecture: We intentionally use question-only hidden states to enable pre-generation abstention. Methods that rely on answer entropy or decoding statistics (like Semantic Entropy) require generating full sequences (often multiple times), increasing inference cost by 5x-10x. SelectLLM’s selection head (a lightweight 2-layer MLP) operates on the prompt's pre-fill state, adding <0.04% latency and allowing the system to "early exit" before generating a single token, saving massive compute.
>
> **Theme 4: Experimental Details, Hyperparameters**
> Questions addressed:
> *R atUF Q2:* Consistency of coverage definitions (theoretical vs. empirical).
> *R DpC2 Q5:* Purpose of expected coverage parameter.
> *R KetN Q1 & Q2 & Q3:* Missing batch size, LR, epochs, and ablation on them.
> *R KetN Q4 & R tA2u Q3:* Sensitivity to target coverage rates and regularization $\lambda$.
> *R tA2u Q5:* Sensitivity to loss weighting $\alpha$ (0.5).
> *R KetN Q7:* Comparison to LoRA calibration (LACIE).
> *R DpC2 Q6:* Report TP/TN ratios.
>
> **Response:**
> Reproducibility: We have added all missing details to the Appendix. The batch size is 120 (simulated via accumulation), and the optimal LR is $1e-6$. We plot Risk-Coverage curves in the revised Appendix showing performance is robust across batch sizes ($30-240$) and target coverages ($c \in \{0.65...0.85\}$).
>
> Baselines: We compared against LACIE (LoRA fine-tuning for calibration) using identical data/compute. SelectLLM outperforms LACIE (TRUTH 752.0 vs 579.3) because LACIE relies on "verbalized" uncertainty, which conflicts with generation quality. In contrast, our decoupled selection head provides precise confidence signals and is jointly trained with the LLM backbone to improve risk-coverage trade-offs.
>
> **Theme 5: Presentation & Citations**
> Questions addressed:
> *R KetN Q8 & R tA2u W1 & R DpC2 Q8:* Missing citations (e.g., for base LLMs, MARS).
> *R tA2u Q1:* Terminology ("Reward function").
>
> **Response:**
> We have corrected the terminology (renamed "reward" to "margin function"), added the missing ablation table (Table 6), and included all requested citations for Llama-3.1, Mistral, Qwen, DeepSeek, and relevant prior work on adjustment to the DPO loss function.

---

### Meta-Review · Area_Chair_W4FS · 2026-01-08

**Summary:**

The paper proposes SelectLLM, designed to abstain with an augmented selection head of the LLM that outputs a confidence score given the question. It finetunes the model by modifying a DPO-style preference loss---forcing the selection probability to be centered around the preferences---and including a coverage loss.

The reviewers' verdicts diverge. On one hand, the reviewers agree with the motivation of the paper. On the other hand, some reviewers question the technical novelties. Perhaps one way to mitigate this question is to evaluate on other calibration metrics that are more direct measurements of the calibration errors, such as ECE, on the model after DPO, and compare with the calibration error of SelectLLM. If the error decreases more significantly, then it is clear that DPO does not guarantee calibration, but the modified empirical selective loss does. From the current results, it is hard to tell how much efficacy the modified loss provides, since there is this risk-coverage tradeoff, and the numbers in the ablation study in the appendix are not that pronounced.

Given the number of negative reviews, I feel that the paper can benefit from another round of thorough revision before its resubmission.

**Reviewer Concerns:**

The novelty and independent validation of confidence remain concerns from the reviews.

**Reviewer Scores:**

I think some of the minor points such as motivation of the modified empirical selective loss would have been addressed, not enough to flip the reviews though.

---

### Decision · Program_Chairs · 2026-01-26

Reject